# Disentangling the Hydrological and Hydraulic Controls on Streamflow Variability in E3SM V2 – A Case Study in the Pantanal Region

Donghui Xu[1*], Gautam Bisht[1], Zeli Tan[1], Chang Liao[1], Tian Zhou[1], Hong-Yi Li[2], Lai-yung Ruby Leung[1]

[1]Atmospheric, Climate, & Earth Sciences Division, Pacific Northwest National Laboratory, Richland, WA, USA
[2]Department of Civil and Environmental Engineering, University of Houston, Houston, Texas, USA

*Correspondence to*: Donghui Xu (donghui.xu@pnnl.gov)

**Abstract**. Streamflow variability plays a crucial role in shaping the dynamics and sustainability of Earth's ecosystems, which can be simulated and projected by river routing model coupled with land surface model. However, the simulation of streamflow at large scales is subject to considerable uncertainties, primarily arising from two related processes: runoff generation (hydrological process) and river routing (hydraulic process). While both processes have impacts on streamflow variability, previous studies only calibrated one of the two processes to reduce biases in the simulated streamflow. Calibration focusing only on one process can result in unrealistic parameter values to compensate for the bias resulted from the other process, thus other water related variables remain poorly simulated. In this study, we performed several experiments with the land and river components of Energy Exascale Earth System Model (E3SM) over the Pantanal region to disentangle the hydrological and hydraulic controls on streamflow variability in coupled land-river simulation. Our results show that the generation of subsurface runoff is the most important factor for streamflow variability contributed by runoff generation process, while floodplain storage effect and main channel roughness have significant impacts on streamflow variability through the river routing process. We further propose a two-step procedure to robustly calibrate the two processes together. The impacts of runoff generation and river routing on streamflow are appropriately addressed with the two-step calibration, which may be adopted by Land Surface Model and Earth System Model developers to improve modelling of streamflow.

## 1 Introduction

Streamflow represents a critical component in the water cycle and an essential freshwater resource to humanity. As the response of the land surface to atmosphere forcings (i.e., precipitation, temperature, radiation, etc.), streamflow exhibits strong seasonality and annual variability that varies regionally (Dettinger and Diaz, 2000). It is vital to understand streamflow variation for any region, since it has critical impacts on water management (Dobriyal et al., 2017), irrigation (Slater and Villarini, 2017), flooding control (Xu et al., 2021), and ecosystem services (Knight et al., 2014). As the hydrological cycle is

intensified by global warming, streamflow characteristics (Xu et al., 2021; Milly et al., 2005; Gudmundsson et al., 2021; Hirabayashi et al., 2013; Bloschl et al., 2017) such as magnitude, seasonality, and frequency may also be modulated. Robust predictions of streamflow variation are crucial for adapting to the consequences of global warming in the future.

Land surface models (LSMs) coupled with river transport models (RTMs) and fully coupled Earth system models (ESMs) have been used to predict the variability of streamflow at large scales to assess water availability and flood risk

(Hirabayashi et al., 2013; Milly et al., 2002; Schewe et al., 2014). However, it remains challenging for the current generation of large-scale models to capture the streamflow seasonality accurately (Xu et al., 2021; Zhang et al., 2016). In addition, the sensitivity of streamflow to climate change is not well represented in existing model simulations (Lehner et al., 2019; Xu et al., 2022a), resulting in inevitable low confidence in the future projections. Although multiple downscaling approaches that use observations as constraints have been applied to reduce model biases in their future projections (Knutti et al., 2017; Lehner

et al., 2019; Tebaldi et al., 2005; Yang et al., 2017), the corresponding uncertainty may not be constrained appropriately due to the resolution mismatch between observation and simulations (Xu et al., 2019; Smith et al., 2009). Therefore, it is necessary to improve the model performance before any reliable conclusions on future projections can be made. The most common way of improving the performance is to calibrate model parameters, which contribute an important source of model uncertainty (Ricciuto et al., 2018; Qian et al., 2018; Qian et al., 2016; Cheng et al., 2021).

The uncertainties of simulated streamflow variability stem from two major natural processes: runoff generation and river routing. Specifically, runoff is first generated in the LSMs (or land component in ESM), whose variability is controlled by hydrological processes, such as infiltration, evapotranspiration, wetland inundation, and soil water dynamics. TOPMODEL (Beven and Kirkby, 1979; Niu et al., 2005), and Variable Infiltration Capacity (VIC; Liang et al., 1994) are the two most widely used runoff generation parameterizations (Sheng et al., 2017). It has been demonstrated that calibrating relevant

parameters in LSMs leads to improved performance in the simulated runoff at site level (Denager et al., 2022), at the watershed scale (Hou et al., 2012; Huang et al., 2013; Liao and Zhuang, 2017), continent scale (Troy et al., 2008; Yang et al., 2019), and global scale (Yang et al., 2021; Xu et al., 2022a). The simulated runoff is then routed as streamflow to the outlet through the river network in RTMs (or river component in ESM). Global RTMs routinely solve kinematic or diffusion-wave approximations of 1D Saint-Venant equations to achieve computational efficiency (Shaad, 2018). Although the physical

process of fluid motion is simplified, the performance of river routing is acceptable in large basins at the monthly time steps, particularly after considering water management effects (Li et al., 2015; Yamazaki et al., 2011; Zhou et al., 2020). Topographic characteristics and channel geometry parameters, which can be derived from finer-resolution topography data, have significant impacts on the simulated hydrograph (Wu et al., 2011; Yamazaki et al., 2009). However, those parameters still need to be calibrated to improve routing process description (Hirpa et al., 2018; Jiang et al., 2021; Xu et al., 2022b) due to resolution

mismatch between model and river networks in the real world (Liao et al., 2022).

Although parameters from both the runoff generation (i.e., hydrological control) and river routing processes (i.e., hydraulic control) significantly influence the streamflow variability, previous studies only focus on one of the processes for

calibration (Hirpa et al., 2018; Huang et al., 2013; Yang et al., 2019; Yang et al., 2021; Mao et al., 2019; Xu et al., 2022b). Calibrating only one process can result in unrealistic parameters to compensate for bias resulted from the other process. Furthermore, a comprehensive calibration needs to consider both processes together, requiring a better understanding of the separate controls from these two processes on streamflow variability. In this study, we aim to disentangle the hydrological and hydraulic controls on the streamflow variability in the coupled land-river configuration of the Energy Exascale Earth System Model (E3SM; Golaz et al., 2019), a fully coupled ESM. The Pantanal region is selected as our study domain because it is challenging for coupled land-river model to simulate its streamflow variability accurately (Schrapffer et al., 2020; Bravo et al., 2012). Specifically, the streamflow seasonality of the Pantanal region is much delayed relative to the precipitation seasonality (e.g., ~5 months). Schrapffer et al. (2020) found the floodplain plain storage effects explain the 5 months delay, but the impacts of hydrological processes were not investigated, making it unclear how the delay is attributed to runoff generation and river routing processes. The significant delay in streamflow seasonality makes Pantanal region an ideal test domain to investigate the separate impacts of hydrological and hydraulic processes on streamflow variability in models. This is because a general watershed has streamflow seasonality closer to precipitation seasonality, hence the uncertainty from atmosphere forcings can be more significant than that from parameterizations. In Sec 2, we briefly introduce the model configuration, study domain, experiment designs, and calibration procedure. We also introduce a modified wetland inundation scheme to improve the representation of wetland inundation process in our model. The results of experimental simulations are first presented in Sec 3, followed by validation of our two-step calibration results against multiple reference datasets. Sec 4 concludes the study.

## 2 Methods

### 2.1 Model description

In this study, we ran the E3SM Land Model (ELM) coupled with MOSART, the river component of E3SM, to investigate the control factors of streamflow variability in coupled land-river model. ELM was developed based on Community Land Model 4.5 (CLM4.5; Oleson et al., 2013), and the same parameterizations of canopy water, snow, runoff generation, and soil water dynamics were used. MOSART is a physically based river routing model that has been coupled with ELM to simulate water transport, including hillslope routing, subnetwork routing, and main channel routing (Li et al., 2013). Luo et al. (2017) coupled a macro floodplain inundation scheme with MOSART to simulate riverine inundation process, which is a necessary process to improve the river model performance (Yamazaki et al., 2011; Decharme et al., 2012; Schrapffer et al., 2020). Water management also plays a crucial role in shaping the streamflow variability (Voisin et al., 2013), but it doesn't have a significant impacts in Pantanal region (Jardim et al., 2020).

### 2.2 Study domain

The Pantanal region, located in the upper Paraguay river basin (Figure 1a), is the world's largest wetland (Ivory et al., 2019; Erwin, 2009). The wetland region may be formed due to the low-lying surface elevation, surrounded by mountains (Figure 1b). The precipitation of this region has strong seasonality, high in January and February and low during June, July, and August (Figure 1c). However, the streamflow at the outlet shows a shift of about 5~6 months in the seasonality compared to precipitation (Figure 1c). The time shift between the streamflow and precipitation is more significant in the downstream subbasins (e.g., SB#4, #5, and #6) than the headwater subbasins (e.g., SB#1, #2, and #3) (Figure S1). While the travel distance of runoff in the headwater subbasins is not very long, the streamflow seasonality is delayed by up to 3 months relative to the precipitation seasonality (Figure S1c and d). We hypothesize the significant delayed response of streamflow to precipitation attributes to both hydrological and hydraulic processes.

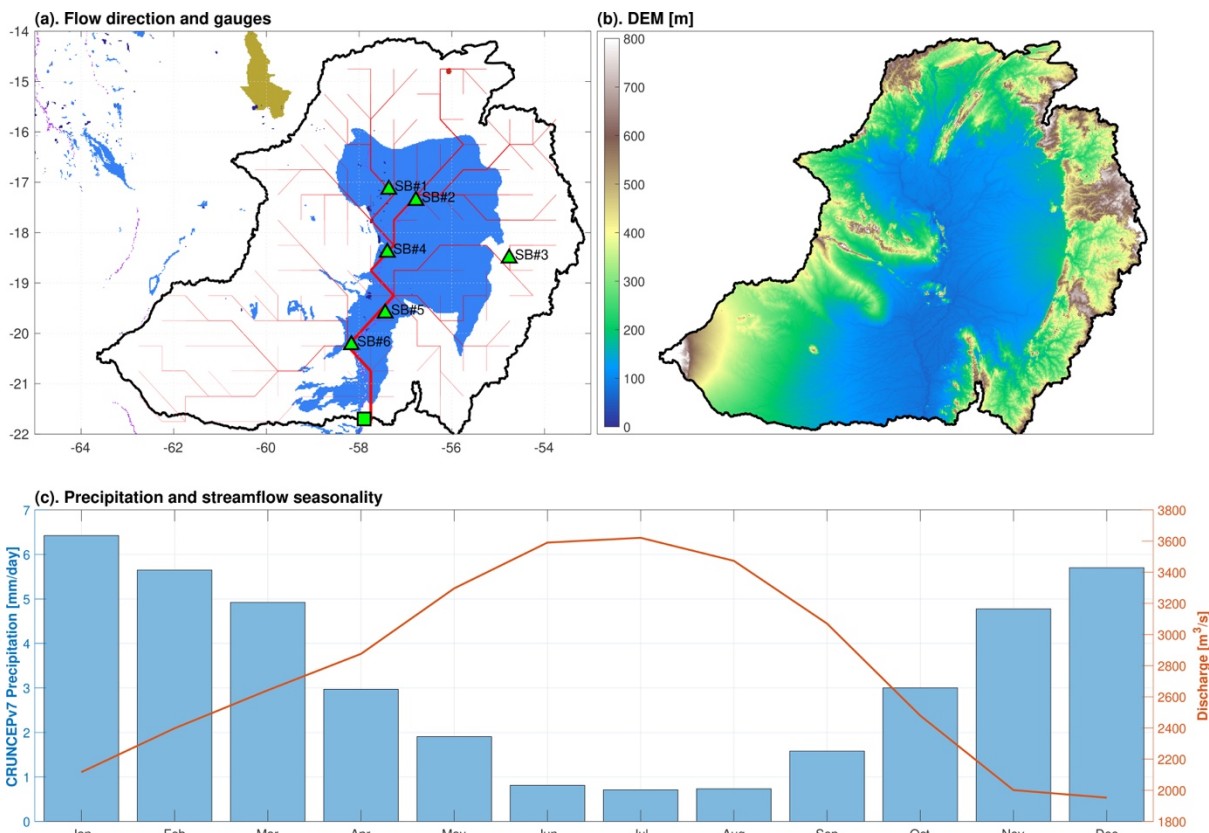

**Figure 1.** (a). Study domain of upper Paraguay basin. The red line shows the river network, and the green square denotes the streamflow outlet (i.e., station Porto Murtinho) used in this study. The green triangles are the sub-basin (SB) gauges that used for evaluation. The blue area represents floodplain according to Global Lakes and Wetlands Database of Lehner and Döll (2004); (b). DEM at 90m resolution of Hydrological Data and Maps Based on Shuttle Elevation Derivatives at Multiple Scales (HydroSHEDS; Lehner et al., 2008); and (c). Precipitation seasonality derived from CRUNCEPv7 (bar plot relies on left Y-

Axis) and observed streamflow seasonality from the outlet of upper Paraguay basin (red solid line relies on right Y-Axis) during 1979-2009.

## 2.3 Model setup

We ran one-way coupled ELM-MOSART (i.e., runoff simulated by ELM is send to MOSART for routing) simulations at a spatial resolution of 0.5° × 0.5° for 1979-2009. The simulations were forced by CRUNCEPv7 atmosphere forcing in this study since it was found to be most accurate over the Pantanal region (Schrapffer et al., 2020). CRUNCEPv7 is a 6-hourly 0.5° × 0.5° global forcing dataset that was generated based on Climatic Research Unit Time-Series Version 3.24 (CRU TS v3.24; Harris et al., 2014) and NCEP reanalysis (Kalnay et al., 1996). The time step for ELM and MOSART is 30 [min] and 60 [min], respectively, with a coupling frequency of 180 [min].

We performed 9 experiments as listed in Table 1 to investigate the sensitivity of streamflow to different processes. The first experiment is the control simulation, using the default ELM surface data and parameters and the default MOSART parameters. In experiment 2, floodplain inundation was turned on to show the impact of floodplain water storage on the streamflow seasonality. Experiment 3 tested the uncertainty of river geometry, and Experiment 4 tested the uncertainty of main channel Manning coefficient ($n_r$). In Experiment 5, we perturb the decay factor for the subsurface runoff generation ($f_{drai}$), which has been identified as the most sensitive parameter for subsurface water dynamics (Huang et al., 2013; Bisht et al., 2018). In experiment 6, we aim to understand how streamflow is affected by wetland inundation process, for which $f_{over}$ and $f_c$ are critical parameters (See Text S1). Experiment 7 represents calibration of river routing processing, including both geometry and Manning coefficient as uncertain. Experiment 8 calibrates both subsurface and surface water dynamics processes in runoff generation process. Lastly, experiment 9 is the proposed two-step calibration with all the parameters from experiments 3-6 included (details can be found in Sec 2.5). We used the diffusion wave routing in MOSART for all the experiments to include backwater effects.

River geometry is assumed to be rectangular (Figure 1a of Luo et al. (2017)), and the main channel bankfull width ($w$), and depth ($d$) can be derived with the equations proposed by Andreadis et al. (2013):

$$w = a_w Q^{0.5}, \qquad \text{Eq (1)}$$
$$d = a_d Q^{0.3}, \qquad \text{Eq (2)}$$

where $Q$ is the 2-year return period daily streamflow, which is estimated at each grid cell by aggregating daily runoff of Reach-level Flood Reanalysis dataset (GRFR; Yang et al., 2021) from the corresponding upstream area, and $a_w$ and $a_d$ are curve fitting parameters. Andreadis et al. (2013) found the 95% confidence intervals for $a_w$ and $a_d$ are $[2.6, 20.2]$ and $[0.12, 0.63]$, respectively. Subsurface runoff ($R_{drai}$) is parameterized as exponential function of $f_{drai}$ and water table depth ($z_\nabla$):

$$R_{drai} = q_{drai,max} \exp(-f_{drai} z_\nabla), \qquad \text{Eq (3)}$$

where $q_{drai,max}$ is the maximum drainage rate. A surface water storage (i.e., depression areas where excess runoff accumulates) (Ekici et al., 2019) was introduced in ELM to simulate wetlands and sub-grid water bodies, and details can be found in Text S1. A modified wetland scheme of Xu et al. (2023b) is adopted here to improve the wetland inundation simulation (see details in Text S1), and two parameters were found to be sensitive for the wetland inundation: $f_{over}$ and $f_c$.

In summary, $a_w$, $a_d$, and $n_r$ are the parameters in MOSART and experiment 2, 3, 4, and 7 are hydraulic sensitivity 140 experiments. $f_{drai}$, $f_{over}$, and $f_c$ are the parameters in ELM and experiment 5, 6, and 8 are hydrology sensitivity experiments.

**Table 1. Simulation experiments design**

| # | Experiment | Inundation mode | Parameters | Objective | Number of simulations |
|---|---|---|---|---|---|
| 1 | Default | off | Default | Streamflow seasonality | 1 |
| 2 | Inundation | on | Default | Streamflow seasonality | 1 |
| 3 | Geometry | on | $a_w \sim U(2.6, 20.2)$ <br> $a_d \sim U(0.12, 0.63)$ | Streamflow seasonality | 100 |
| 4 | Manning coefficient | on | $n_r \sim U(0.01, 0.2)$ | Streamflow seasonality | 100 |
| 5 | Subsurface runoff | on | $f_{drai} \sim U(0.1, 5)$ | Streamflow seasonality | 100 |
| 6 | Wetland inundation | on | $f_{over} \sim U(0.1, 5)$ <br> $\log(f_c) \sim U(1e^{-3}, 1e^{-0.155})$ | Streamflow seasonality | 100 |
| 7 | MOSART calibration | on | $a_w, a_d, n_r$ | Streamflow seasonality | 1,000 |
| 8 | ELM calibration | on | $f_{drai}, f_{over}, f_c$ | Streamflow seasonality | 1,000 |

| 9 | Two-step calibration | on | $a_w, a_d, n_r, f_{drai}, f_{over}, f_c$ | Multi-objectives | 2,000 |
|---|---|---|---|---|---|

## 2.4 Data

Multiple datasets of site observations, satellite observations, and reanalysis products were used in this study for model calibration and evaluation. The simulated streamflow was validated at monthly and annual time scales. We used monthly streamflow observation at gauge Porto Murtinho (basin outlet shown in Figure 1a) and 6 internal subbasin gauges (Figure 1a) that archived in Global Stream Indices and Metadata (GSIM; Do et al., 2018; Gudmundsson et al., 2018). The three global runoff datasets that were used to benchmark the performance of annual streamflow include Global Runoff Reconstruction (GRUN; Ghiggi et al., 2019), Linear Optimal Runoff Aggregate (LORA; Hobeichi et al., 2019), and Global Reach-level Flood Reanalysis (GRFR; Yang et al., 2021). The global surface water dynamics dataset from Global Land Analysis & Discovery (GLAD; Pickens et al., 2020) was used to validate the simulated surface water fraction (SWF), which is the sum of ELM-simulated inundation and MOSART-simulated inundation in E3SM (Xu et al., 2023b). GLAD provides global monthly, annual, and seasonal surface water and permanent water layer derived from Landsat images during 1999-2020 at $0.00025° \times 0.00025°$ ($\sim 30m \times 30m$). We used seasonal GLAD and removed the permanent water to exclude the rivers and lakes from the SWF. The GLAD was upscaled to $0.5° \times 0.5°$ by averaging the values from all the finer resolution grid cells within the coarse model resolution grid cell to compare with the model simulations. The gridded energy flux data (FLUXCOM; Jung et al., 2019) and Global Land Evaporation Amsterdam Model version 3 (GLEAMv3; Martens et al., 2017) product were used to evaluate evapotranspiration (ET).

## 2.5 Calibration procedure

We propose a two-step calibration procedure in this study to appropriately resolve hydrological and hydraulic impacts on streamflow variability. In step 1, we run 1,000 simulations with runoff generation-related parameters ($f_{drain}$, $f_{over}$, and $f_c$) randomly sampled from the prior distributions given by experiment 8 in Table 1. A multi-objective function was proposed by Yang et al. (2019) to calibrate a land surface model, which is adopted in this study. The best parameter was calibrated in each individual grid cell to minimize the following objective function ($obj$) to capture SWF, baseflow index (BFI), and annual streamflow trend (Trend) simultaneously:

$$obj = w_1 \cdot \left| \log\left(\frac{SWF_{sim}}{SWF_{glad}}\right) \right| + w_2 \cdot \left| \log\left(\frac{BFI_{sim}}{BFI_{GSCD}}\right) \right| + w_3 \cdot \left| \log\left(\frac{Trend_{sim}}{Trend_{obs}}\right) \right|, \qquad \text{Eq (4)}$$

Where $SWF_{sim}$ represents annual averaged simulated surface water fraction from coupled ELM-MOSART simulation, $SWF_{glad}$ is the GLAD annual averaged surface water fraction. $SWF$ was estimated between 1999-2009, when the simulation

period overlapped with GLAD temporal availability. We note the simulated SWF includes both ELM-simulated and MOSART-simulated inundation. But only the ELM relevant parameters were calibrated to optimize Eq (4), as routing process is not considered in step 1 calibration. Therefore, the SWF in the objective function Eq (4) is only affected by ELM processes. $BFI_{sim}$ and $BFI_{GSCD}$ are the baseflow index (i.e., the ratio of subsurface runoff to total runoff) from simulations and Global Streamflow Characteristics Dataset (GSCD; Beck et al., 2013), respectively. $Trend$ denotes the trend of annual runoff quantified by Sen's slope (Sen, 1968). Because there is no reliable gridded dataset of long-term runoff trend, we assumed $Trend_{obs}$ to be constant over the whole basin, and the value was derived from observed streamflow at the outlet. This assumption may introduce uncertainty in the model for capturing the heterogeneity of runoff sensitivity to climate change. Although there exist several internal sub-basin gauges, their temporal coverages are shorter than the simulation period, so they cannot be used to derive annual runoff trends within the watershed. Due to the different bias magnitude and uncertainty pattern, the assignment of weights is subjective (Yang et al., 2019). In step 2, another 1,000 simulations were performed with river routing parameters ($a_w$, $a_d$, and $n_r$) randomly sampled from the uniform distribution in Table 1 and the calibrated hydrological parameters from step 1. The best $a_w$, $a_d$, and $n_r$ were calibrated at basin level by maximizing the correlation coefficient between monthly simulated and observed streamflow at outlet. As suggested by Shen et al. (2022), we used all the streamflow observation for calibration without splitting to validation period. We further validate the calibrated model at six internal sub-basin gauges (Figure 1a), which are not used during the calibration process.

## 3 Results and discussion

### 3.1 Streamflow variability in Pantanal region

Figure 2 illustrates the observed streamflow seasonality from 6 selected sub-basins in Pantanal region, with the streamflow peaks shift from March in the headwater (sub-basin #1, #2, and #3) to June in downstream of the river (sub-basin #6). Streamflow variability reflects the combined effects of routing surface runoff and subsurface runoff. Surface runoff is closely related to the surface water dynamics, which is controlled by the hydraulic factor, such as floodplain inundation process. However, the surface water dynamics has a much earlier seasonality (i.e., peak in March) than the streamflow at the downstream gauges (Subbasin #4, #5, and #6). The differences in the seasonality between surface water dynamics and outlet streamflow imply the significant contribution of subsurface runoff to the streamflow variability. The subsurface runoff generation is hydrological process, simulated by land surface model (e.g., ELM). Therefore, both hydrological and hydraulic processes have impacts on the significant shift in the seasonality between precipitation and streamflow in Pantanal region.

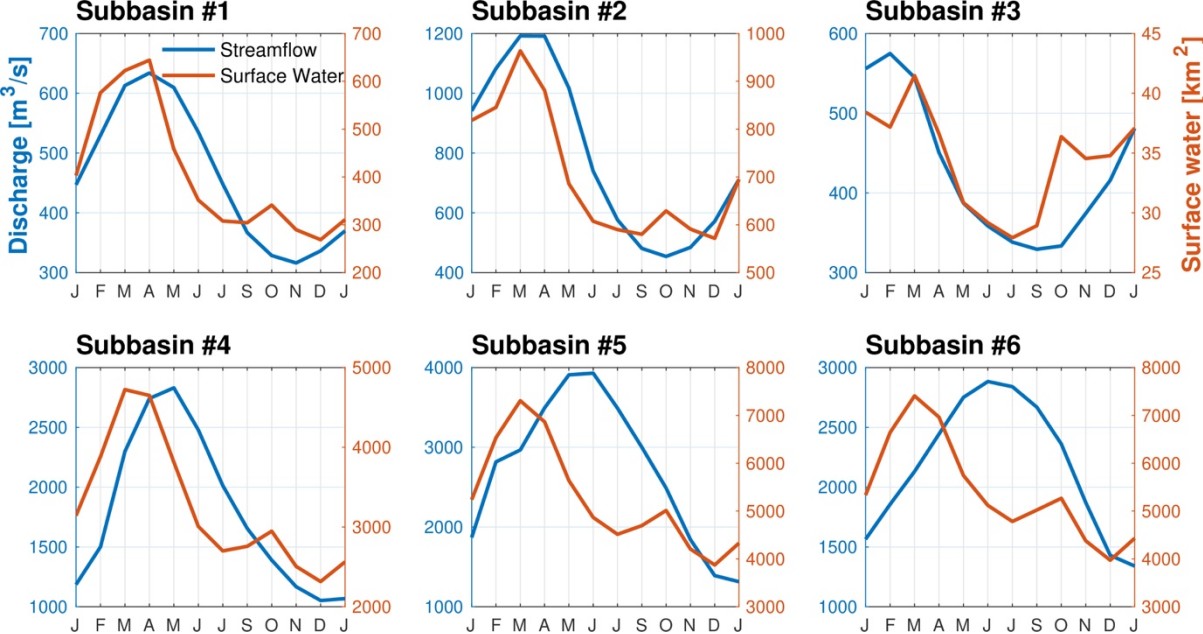

**Figure 2**. Streamflow seasonality derived from GSIM observation during 1979-2009 (blue solid line) and seasonality of surface water dynamic from GLAD dataset (red soild line) at sub-basins in Pantanal region.

## 3.2 Impacts of MOSART on streamflow variability

The default coupled ELM-MOSART simulation (i.e., experiment 1) fails to capture the streamflow seasonality over the Pantanal region. Specifically, the default simulation shows streamflow peaks around March, which is much earlier than the observed streamflow peaks in July (Figure 3). Previous studies found that the late responses of streamflow peaks to

205 precipitation are caused by the storage effects of floodplain (Schrapffer et al., 2020). However, in our inundation experiment (i.e., experiment 2), turning on the inundation mode only delays the streamflow seasonality by one month compared to the default configuration, which still cannot capture the observed seasonality well (Figure 3). This discrepancy suggests significant uncertainties exist in other river routing or runoff generation-related parameters. For example, our geometry experiment (i.e., experiment 3) demonstrates that smaller $a_w$ and $a_d$ lead to a higher model performance of capturing streamflow seasonality,

such as a higher correlation coefficient of monthly streamflow between simulation and observation (Figure 4a). Smaller $a_w$ and $a_d$ (i.e., smaller cross-section area) correspond less channel capacity, implying that more water will inundate on floodplain during flooding period, therefore, the peak streamflow is delayed to a later time. Although the streamflow seasonality is improved by reducing channel cross-section area, it results in smaller temporal variations in monthly streamflow, such as the ratio of standard deviation (rSD) between simulation and observation being less than 0.85. The smaller rSD indicates the

regulation of the floodplain is magnified; therefore, it is not reasonable to accept such small channel geometry.

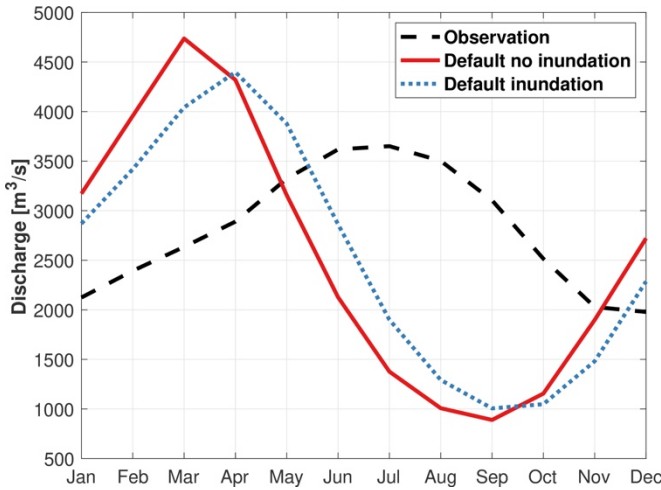

**Figure 3.** Simulated streamflow seasonality of default simulation (experiment 1 in Table 1), and default simulation with inundation mode (experiment 2 in Table 1). The seasonality is derived from 1979 to 2009.

The Manning coefficient of the main channel affects the streamflow seasonality through its impacts on flow velocity. In Manning coefficient experiment (i.e., experiment 4), increasing the channel Manning coefficient will at most shift the occurrence of peak streamflow to June (Figure 4b), while the corresponding value ($nr \geq 0.16$) is much higher than the generally used main channel Manning coefficient in global river transport models (e.g., 0.03-0.06; Yamazaki et al., 2011; Li et al., 2013; Decharme et al., 2012). Larger Manning coefficient means slower flow velocity along the river direction, then the

water will accumulate in the channel and inundate the floodplain when it exceeds the channel capacity. Notably, such high main channel Manning coefficients lead to an unrealistic flatter hydrograph with rSD < 0.7. Using Manning coefficient at around 0.1 reproduces the monthly variability best (e.g., rSD = 1), which is still higher than the typical value, but such a range may be reasonable in this region because of the vegetated surface and complex river network (i.e., meandering) according to Chow (1959). The subnetwork and hillslope manning coefficients in MOSART are not included in our experiments because

they have negligible impacts on the shape of the hydrograph (Figure S4).

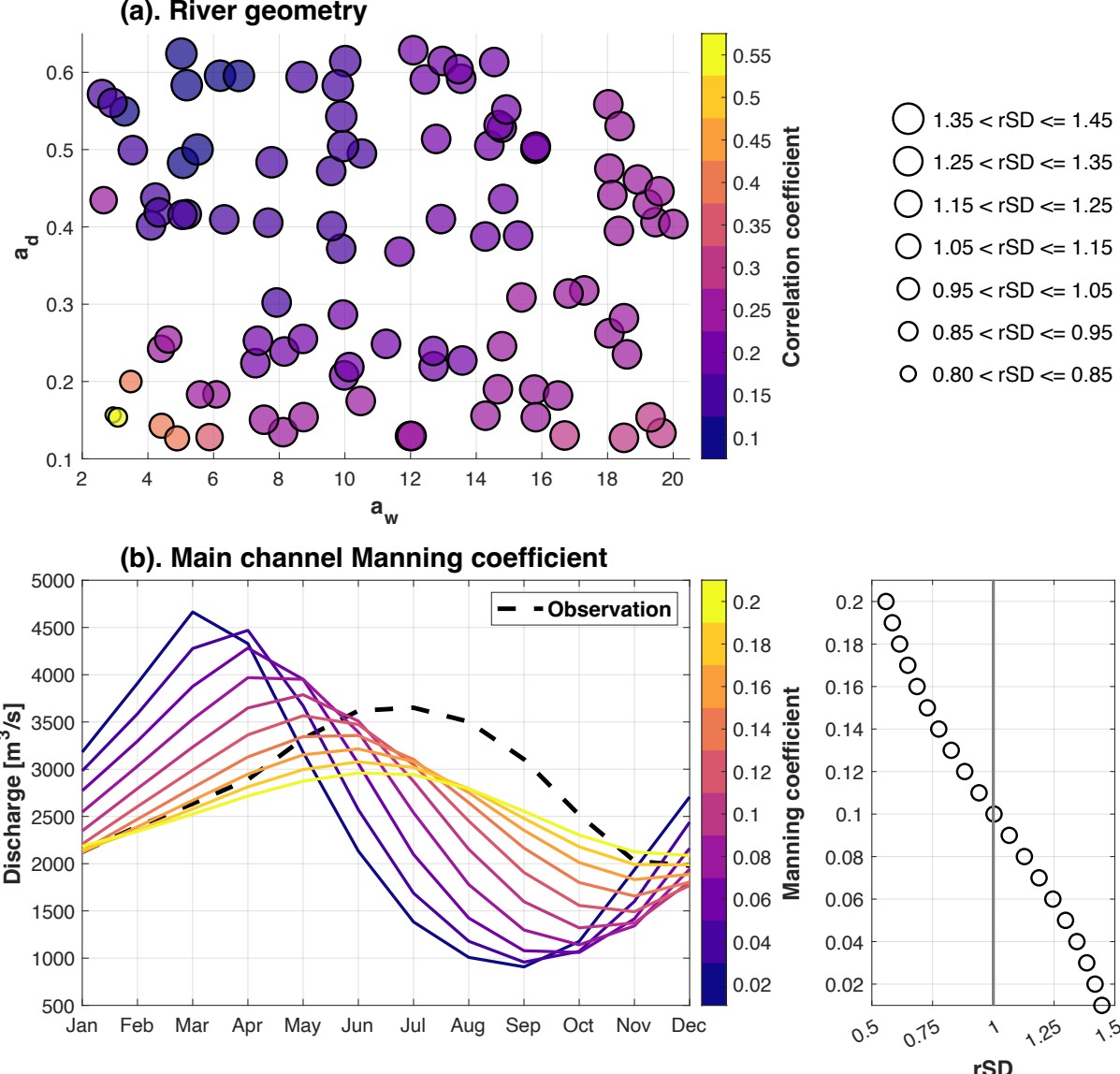

**Figure 4.** Sensitivity of streamflow seasonality and variability to (a). river geometry, and (b) main channel Manning coefficient. In subplot (a), $a_w$ and $a_d$ are the parameters for river width and depth, respectively (Eq (1) and Eq (2)), and the correlation coefficient between the simulated streamflow and observed streamflow at monthly scale is used to indicate the model performance of simulating streamflow seasonality. The scatter size is proportional to the ratio of standard deviation (rSD) between the simulated streamflow and observation. The black dashed line in subplot (b) represents the averaged streamflow seasonality derived from observation. And the scatter on the right shows the relationship between the main channel manning coefficient (Y-axis) and rSD (X-axis).

Although calibrating channel geometry and Manning coefficient improves the performance of the coupled ELM-MOSART model in simulating streamflow, there exist some discrepancies in the streamflow seasonality and variation between simulation and observation. Floodplain effects cannot completely explain the streamflow seasonality in the Pantanal region. Additionally, other variables in the water cycle remain highly biased, which cannot be fixed by calibrating river routing-related parameters. First, the simulated BFI averaged over the basin is about 0.15, which significantly underestimates the values (e.g.,

0.69) estimated by Global Streamflow Characteristics Dataset (GSCD; Beck et al., 2013) (Figure 5a and b). Second, the surface water dynamics is poorly simulated with a low spatial correlation coefficient (0.03) and high percentage bias (821%) as compared to upscaled GLAD dataset (Figure 5c and d). Third, the Pantanal region has become drier in the past several decades (Libonati et al., 2020), which can be detected from the observed annual streamflow time series (Figure 5e). While the model is able to capture the drier trend of annual streamflow during the simulation period (negative Sen's slope with p-value = 0.05),

the magnitude of the decreasing trend is underestimated (Figure 5e). The above discrepancies can be reduced by calibrating the runoff generation related processes (e.g., hydrological control) because river routing process only impact the shape of the hydrograph.

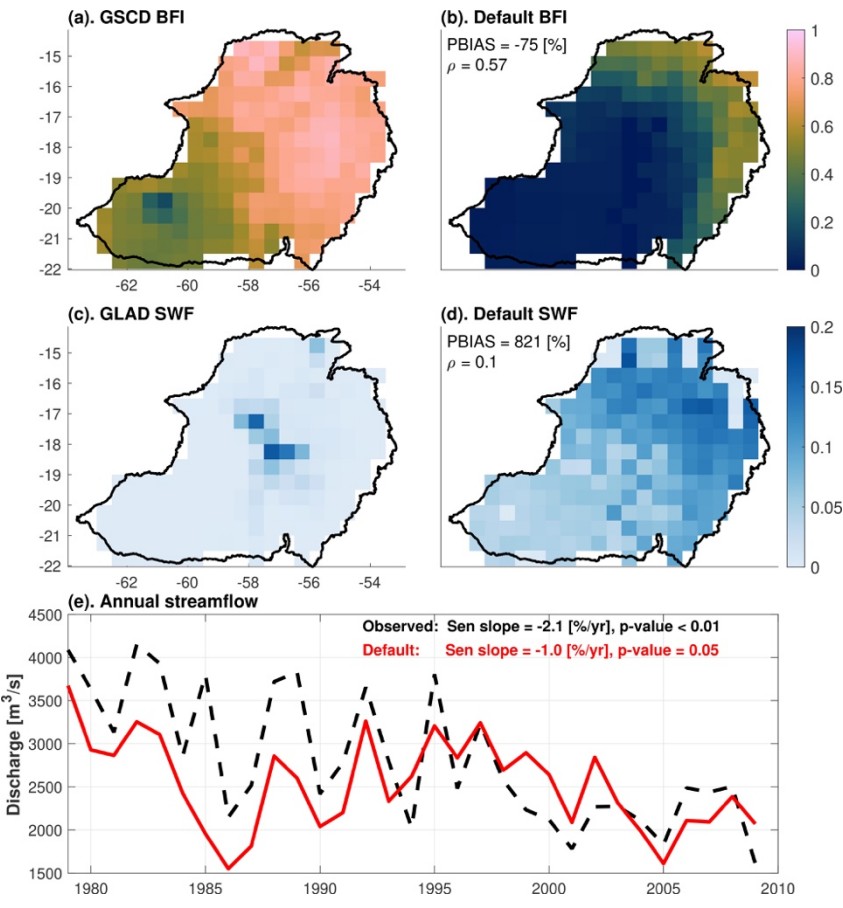

**Figure 5.** Performance of ELM-MOSART with default configuration. (a). Base flow index (BFI) from GSCD BFI3; (b).
simulated BFI with default parameter; (c). surface water fraction (SWF) from GLAD; (d) simulated SWF (floodplain inundation + wetland inundation) with default parameter; (e). annual streamflow time series of observation (black dashed line) and default simulation (red solid line) at basin outlet.

### 3.3 Impacts of ELM on streamflow variability

Subsurface runoff experiment (i.e., experiment 5) demonstrates subsurface runoff generation process has a critical control on the streamflow seasonality as well (Figure 6). It is shown in Figure 6a that the simulated streamflow tends to peak later as $f_{drain}$ decreases. Decrease of $f_{drain}$ means the subsurface storage capacity increases (Huang et al., 2013), therefore, leading to a higher base flow index (Figure 6b). This explains the response of total runoff to rainfall is delayed when smaller $f_{drain}$ was used due to the slower mechanism of subsurface runoff (Vivoni et al., 2007). Furthermore, smaller $f_{drain}$ leads to
increased annual streamflow magnitude but reduced monthly streamflow variability (Figure 6c). We acknowledge unrealistic high streamflow (e.g., mean > 5,000 $[\frac{m^3}{s}]$) with minimal month-to-month variability (e.g., rSD < 0.4) are found in the simulations when $f_{drain}$ is close to 0.1; thus, a more reasonable lower bound of $f_{drain}'s$ prior should be modified to a relatively larger value (e.g., 0.25). Moreover, the trend of annual simulated streamflow can be affected by $f_{drain}$, though all the simulations tend to underestimate the trend detected in the observation. The decreasing trend is more significant and closer to
the observed trend when $f_{drain}$ decreases (Figure 6d), except two problematic simulations with very small $f_{drain}$.

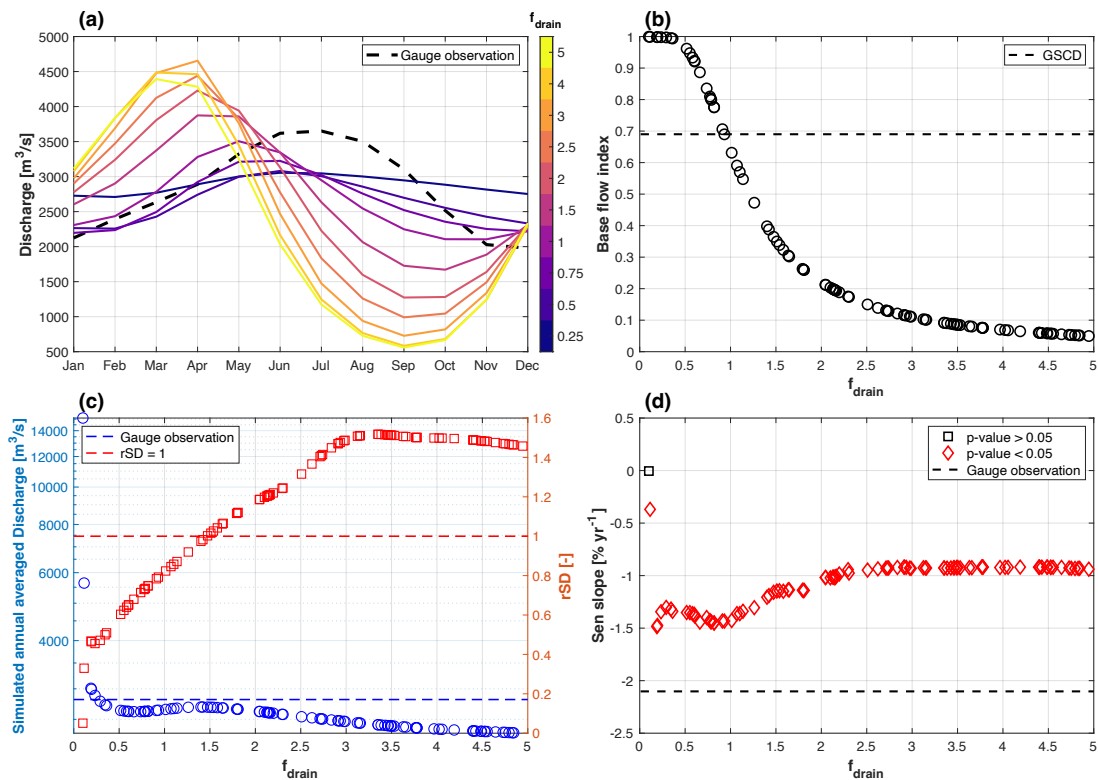

**Figure 6.** Sensitivity of (a). Streamflow seasonality; (b). Base flow index; (c). Annual streamflow mean and rSD; and (d). annual streamflow trend to $f_{drain}$. rSD denotes the ratio of monthly standard deviation between simulation and observation. Note, the Sen's slope is normalized by the annual averaged streamflow.

The sensitivity of annual runoff trend (i.e., annual streamflow trend) to $f_{drain}$ results from the impacts of $f_{drain}$ on ET processes. As a conceptual example, the runoff will not have trend in time if precipitation and ET have similar decreasing trend (e.g., P and $E_1$ in Figure 7a). Runoff will show a trend in time when the changing rate of ET is different from precipitation (e.g., P and $E_2$ in Figure 7a). The subsurface runoff experiment shows that the ET is less sensitive to precipitation decreases

with smaller $f_{drain}$ (Figure 7b), thus leading to a larger decreasing trend in annual runoff (light green line in Figure 7a). Based on above analyses, it is reasonable to argue that a good selection of $f_{drain}$ in this region should be smaller than the default value (i.e., 2.5), which will yield later streamflow seasonality and better estimates of base flow index, monthly variability, and annual trend compared to reference data or observation.

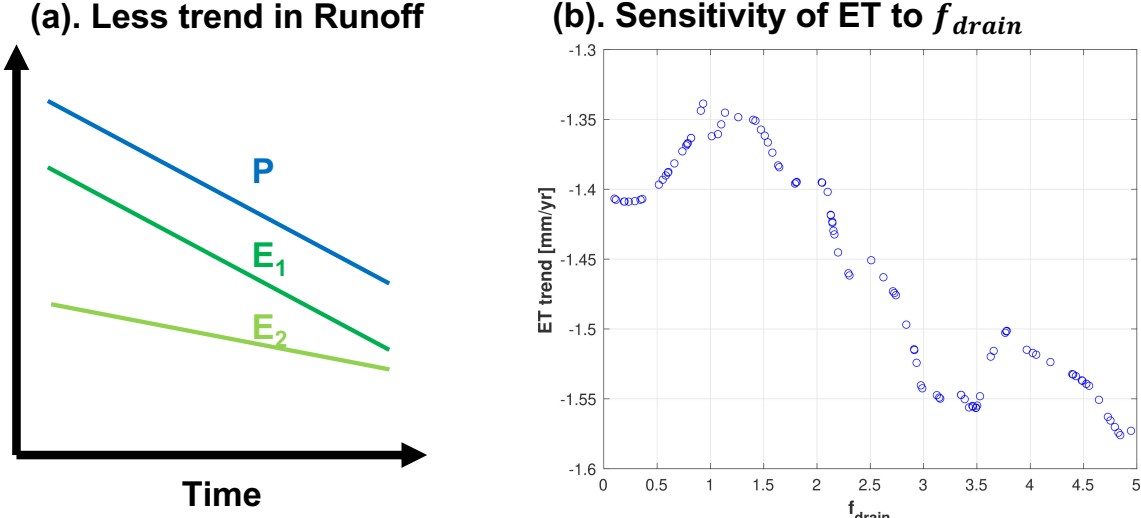

**Figure 7.** (a) Conceptual plot of annual precipitation (P) and annual evapotranspiration (ET) with different trend (E₁ and E₂). Subplot (b) shows the sensitivity of annual ET trend (e.g., Sen's slope of annual basin averaged ET during the simulation period) to $f_{drain}$.

Parameters $f_{over}$ and $f_c$ control the streamflow variability through wetland inundation process based on our wetland inundation experiment (i.e., experiment 6) (Figure 8a) and previous study (Xu et al., 2023b). Wetland inundation is only sensitive to $f_c$ when $f_{over}$ is smaller than 0.5, as small $f_{over}$ leads to a higher saturated fraction, at which infiltration from the wetlands is constrained and inundation occurs (Figure S3a). In contrast to the impacts of floodplain inundation, higher wetland inundation is associated with earlier streamflow peaks (Figure 8b). This is because high wetland inundation is caused by precipitation when the saturation fraction is high (e.g., lower $f_{over}$), consequently, a larger fraction of precipitation converts to surface runoff. Hence, the streamflow seasonality is very close to precipitation seasonality when high wetland inundation is simulated since surface runoff responds quickly to precipitation. The annual streamflow magnitude and trend are also sensitive to the wetland inundation process (Figure 8c and d), especially when parameter $f_{over}$ falls in the lower range (e.g., less than 1.5). Specifically, the annual streamflow magnitude increases, and the trend become less negative as $f_{over}$ decreases.

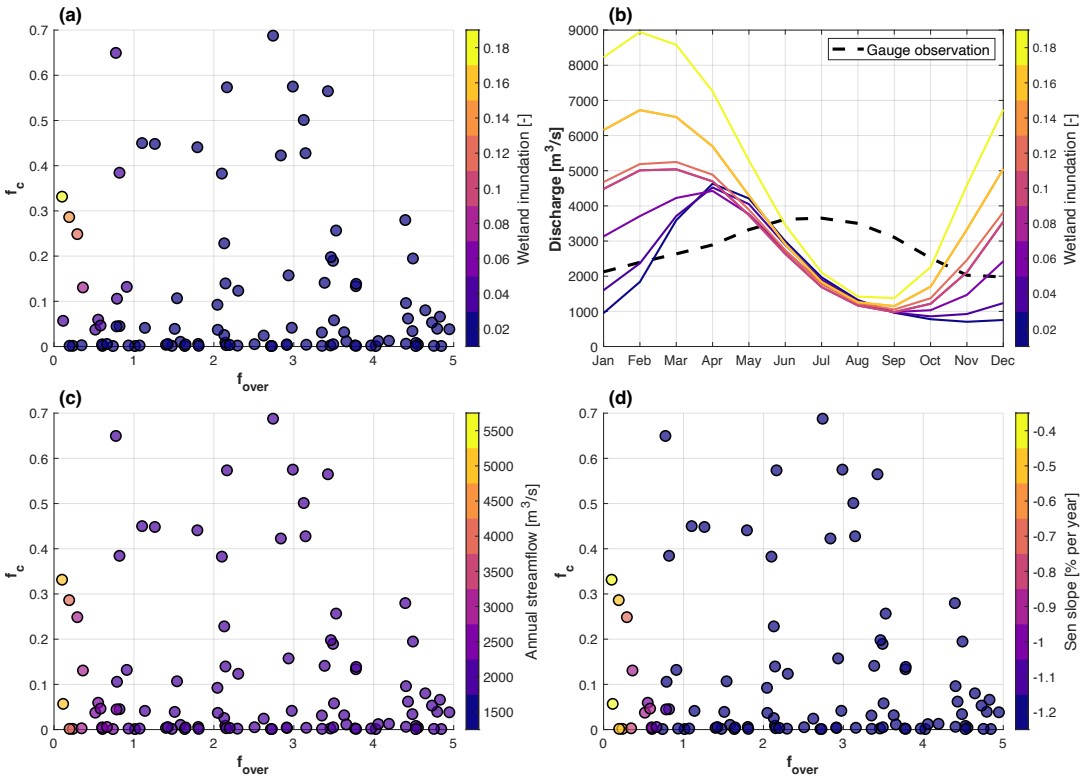

**Figure 8.** Subplot (a) sensitivity of wetland inundation to $f_c$ and $f_{over}$. Subplot (b) impacts of wetland inundation on streamflow seasonality. Sensitivity of (c). annual streamflow; and (d). annual streamflow trend to $f_c$ and $f_{over}$. Note, the Sen's slope is normalized by the annual averaged streamflow.

### 3.4 Single process calibration

We further conducted two independent calibrations for either the river routing process in MOSART and the runoff generation process in ELM using experiments 7 and 8 (Table 1), respectively. Specifically, the objectives of both independent calibrations are to maximize the correlation coefficient between the simulated and observed streamflow time series. Both calibrations can improve the streamflow seasonality significantly as compared to the default configuration, with correlation coefficient increased from 0.02 to 0.75 and 0.67, respectively (Figure 9a and b). The long-term averaged annual streamflow remains similar in the default configuration and two calibrations, with all underestimating the observation by about 10%. However, the temporal variations of simulated streamflow time series in the two calibrations show divergent bias patterns, with MOSART calibration resulting in a smaller standard deviation (e.g., rSD = 0.74), while ELM calibration leading to a slightly higher standard deviation (e.g., rSD = 1.06) (Figure 9b).

Individual MOSART calibration can lead to a very high Manning coefficient (e.g., $nr = 0.17$) and a smaller cross-section of the main channel to capture the streamflow seasonality. Consequently, the flow velocity is slower, and more river

water inundates the floodplain due to backwater effects, which results in a significant underestimation in the temporal variation of simulated streamflow (Figure 9b). The individual ELM calibration leads to an unreasonably higher baseflow index (Figure 9b), implying the subsurface and surface water dynamics are not well represented. Overall, only calibrating the parameters related to one process results in unrealistic parameter values to compensate the bias from the other process. Therefore, the impacts of both runoff generation and river routing processes on streamflow must be considered together in calibrating the

streamflow simulation to ensure all the processes of the water cycle are well calibrated.

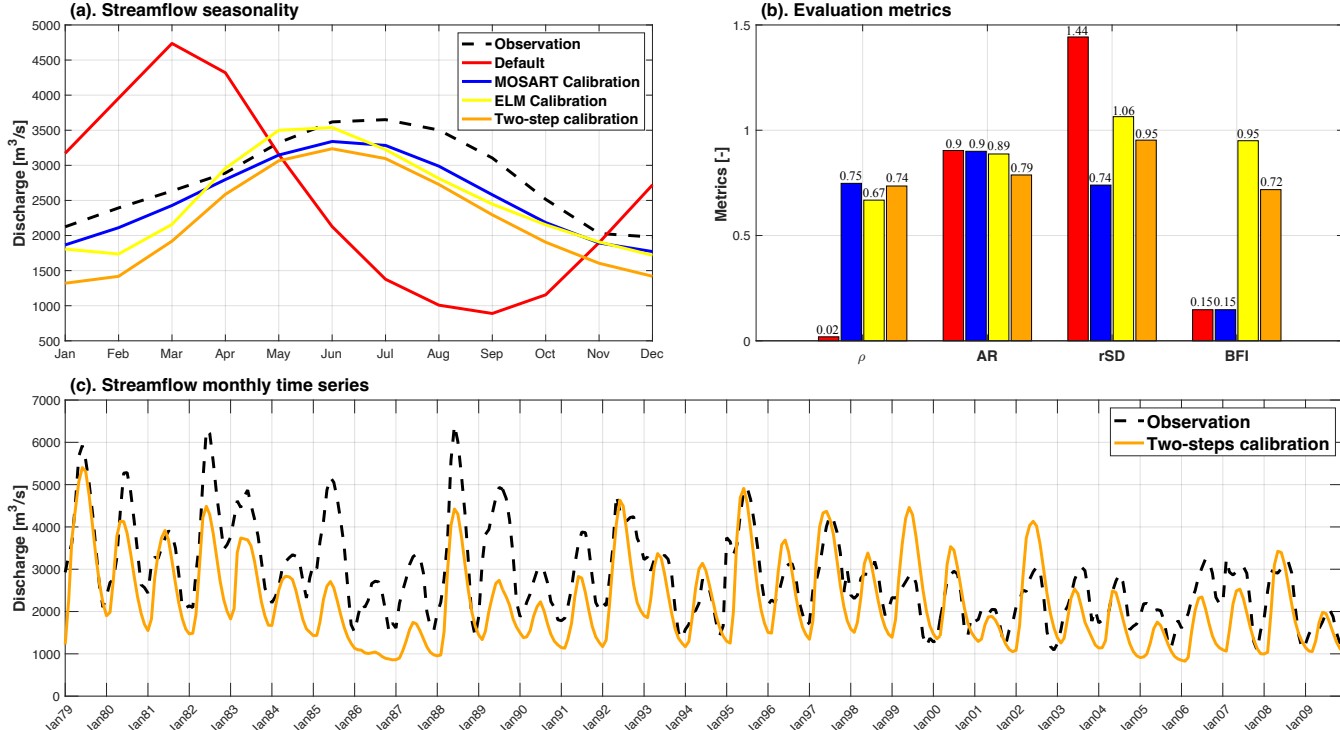

**Figure 9.** Comparison of hydraulic, hydrology, and two-step calibration with default configuration. Subplot (a) shows the seasonality from simulation period, and subplot (b) denotes the correlation coefficient ($\rho$), annual streamflow ratio (AR), ratio of standard deviation between simulation to observation (rSD), and baseflow index (BFI). Subplot (c) presents the monthly

time series of simulated streamflow after two-step calibration.

### 3. 5 Two-step calibration

We propose a two-step calibration procedure in this study (Sec 2.5) by first calibrating the runoff generation relevant parameters in ELM, and then calibrating the river routing relevant parameters in MOSART. Since it is unclear how much the streamflow delay is attributed to the runoff generation process, we use other objectives (i.e., independent of streamflow

variability) in the ELM calibration (Eq (4)). It is implied from the analyses of runoff-related parameters that the runoff seasonality is related to the objectives used in Eq (4). Therefore, we assume the delay of streamflow attributed to the runoff

generation process can be addressed by minimizing Eq (4). In the multi-objective function, we found $w_1 = 0.15$, $w_2 = 0.7$, and $w_3 = 0.15$ yield the best result. The calibrated simulation shows good skill in capturing the spatial pattern of baseflow index and surface water fraction with correlation coefficients of 0.7, and 0.64, respectively, as compared to the reference data

(Figure 10a and b). The absolute biases are significantly reduced as well. Moreover, the calibrated simulation shows a statistically significant decreasing trend in the annual streamflow with Sen's slope equal to -1.80 [%/yr] after normalizing with averaged annual streamflow magnitude. As a benchmark, we found other calibrated or bias-corrected runoff datasets cannot capture either the annual magnitude or annual trend (Figure 10c). The uncertainty of the annual magnitude may stem from the forcing uncertainty, while the trend bias is because model sensitivity to climate change is not well constrained. This suggests

including trend as a performance metric in the objective function is necessary for calibrating ESMs, which are commonly used to project the change of water, energy, and carbon cycle under warmer climate. Although water table depth is not included in the objective function, the simulated water table depth is more consistent with the dataset of Fan et al. (2013) than the default configuration, which simulates a very shallow water table (Figure S5). The improved representation of groundwater table depth is because that the subsurface and surface runoff are correctly separated (e.g., a better simulated baseflow index).

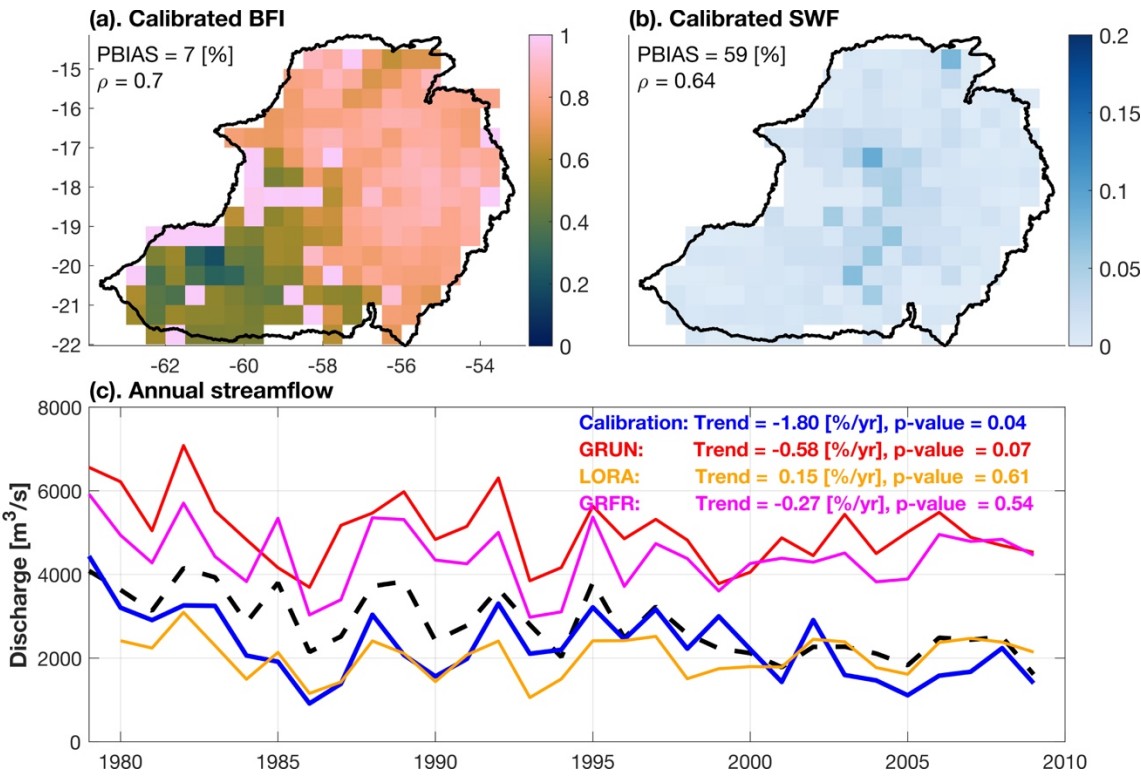

**Figure 10.** Performance of ELM-MOSART with calibrated ELM parameters for (a). Base flow index (BFI); (b). surface water fraction (SWF); and (c). annual streamflow trend. Three other runoff datasets are used to provide benchmarks for annual streamflow. The black dashed line represents the observation from the outlet, with the annual trend (i.e., Sen's slope) to be -2.1 [%/yr] and the corresponding p-value less than 0.05.

With hydrology parameters determined from the step 1 calibration, the model shows good performance of simulating monthly streamflow after the step 2 calibration (Figure 9c). Specifically, the correlation coefficient between simulation and observation at the monthly scale is 0.74, indicating the streamflow seasonality is well captured. Compared to the individual MOSART or ELM calibration, the two-step calibration shows better skill in both monthly variation (e.g., rSD=0.95) and baseflow index (e.g., BFI=0.72) (Figure 9b). However, the two-step calibration exhibits higher biases in the annual streamflow magnitude (e.g., 21% underestimation or 32 [mm/yr] in absolute magnitude), stemming from the following reasons: 1) uncertainty of upscaled flow directions in MOSART because of the relatively coarse spatial resolution ($\sim50km \times 50km$) (Liao et al., 2022). For example, the contributing area of the selected outlet estimated using the modeled flow direction underestimates $\sim5\%$ of the contributing area that is delineated with the 1km HydroSHEDS DEM; 2) uncertainty of forcing data. Specifically, the annual averaged precipitation in CRUNCEPv7 is lower than the mean of multiple precipitation products (Figure S6a). Considering the selected precipitation ensemble (Table S1), the uncertainty of CRUNCEPv7 precipitation varies between -73 – 49 [mm/yr]. There may exist biases in other forcing variables (e.g., humidity and radiation) as well (Figure S7), resulting in an overestimation of evapotranspiration during the dry period (e.g., May to July) as compared to FLUXCOM (Figure 11). For the reference ET of GLEAM dataset, the simulated ET is higher for all the months. The overestimation of simulated annual averaged ET is 54 [mm/yr] and 154 [mm/yr] compared to FLUXCOM and GLEAM, respectively. In addition, Xu et al. (2023a) also reported that energy partition algorithm in current ESMs tends to overestimate ET likely due to the uncertainty in surface condition and parameterization of aerodynamic and stomatal resistance. In summary, while our calibrated model underestimates streamflow magnitude, the result is reasonable as the negative bias is consistent with the smaller contributing area, lower precipitation forcing, and higher evapotranspiration estimates. Furthermore, CRUNCEPv7 shows a notable reduction of precipitation from January to February, while other precipitation products show similar precipitation volume in January and February (Figure S6b). The underestimation of precipitation in February further explains the slightly earlier streamflow seasonality after the two-step calibration.

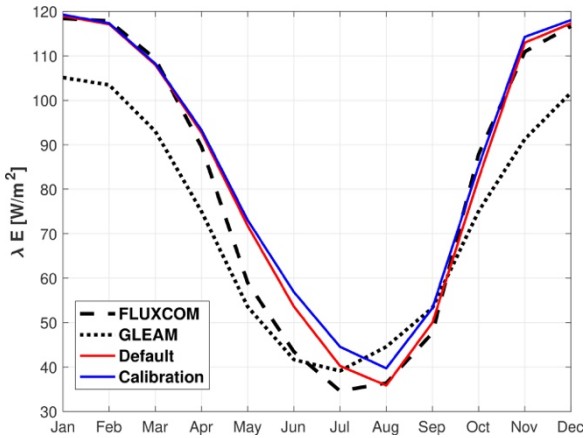

**Figure 11.** Comparison of latent heat flux between simulations and references.

After the two-step calibration, both the runoff generation and river routing processes are robustly improved in E3SM over Pantanal region. For example, the objective function of Eq (4) reduces from 20.8 to 4.1 after step 1 calibration and the correlation between the simulated streamflow and observed streamflow at outlet increases from 0.2 to 0.74 after step 2

calibration. Although only the observed streamflow at the basin outlet was used during the calibration, the performances of simulated streamflow are improved at selected subbasin gauges compared to default configuration (Figure 12). Specifically, compared to the default simulation, the step 1 calibration shifts the streamflow by about 1 month. The step 2 calibration further delays the simulated streamflow seasonality, especially for the downstream subbasins (subbasin #5 and #6). Overall, the calibrated model captures the streamflow monthly variation at all the sub-basin gauges well with correlation coefficient larger

than 0.6. While the seasonality is well captured, the calibrated model cannot reproduce the magnitude of streamflow well at the headwater sub-basins (Subbasin #1, #2, and #3), likely due to the uncertainty of precipitation in the forcing and bias of flow direction represented in MOSART (Figure S8). The relative higher uncertainties in small subbasins are expected in the context of ESM, as large-scale coupled land-river model is commonly evaluated at global major river basins (Yamazaki et al., 2011; Li et al., 2015; Decharme et al., 2019). We note the number of simulations in the two-step calibration, such as a total of

2,000 simulations, is enough to identify the parameter values that approximate the best parameter values. As shown in Figure S9, the improvements of model performance reach a plateau when the number of calibration simulations is larger than 200 in both steps.

Figure S10 shows the calibrated parameter values, and we note it is hard to identify the relationship between the calibrated parameters and watershed characteristics due to the coarse spatial resolution of ESM and its simplifications in

physical process. As process-based models, some parameters in ELM and MOSART are derived from surface and subsurface conditions and properties. For example, in ELM, saturated hydraulic conductivity and specific yield are estimated based on soil types, maximum drainage rate is determined by topographic slope, etc. In MOSART, river length and slope are derived from high resolution DEM. However, some other parameters should be determined based on sensitivity analysis and calibration, such as the parameters selected in this study. This is because ESM resolutions are typically too coarse to represent

some small-scale physical processes, and empirical functions are used to parameterize those processes with simplifications. Specifically, $f_{drain}$ and $f_{over}$ are decay factors of the exponential function in the subsurface and surface runoff generation processes, respectively. According to the development of the runoff scheme in our model (e.g., simple TOPMODEL-based runoff parameterization), $f_{drain}$ and $f_{over}$ should be determined through sensitivity analysis or calibration against hydrograph recession curve (Niu et al., 2007; Niu et al., 2005). $f_c$ is a threshold below which no single connected inundated area spans the

grid cell. It is used to quantify the fraction of the inundated portion of the grid cell that is interconnected according to percolation theory. In other words, $f_c$ determines the maximum inundation extent, above which the water will drain from the inundated area. Although the maximum inundated area should be controlled by topographic variation, high resolution data that

captures the topographic variation under the inundated areas is not available. There exist observations of river width and depth at very high spatial resolution, but it is challenging to upscale the observed river width and depth to ESM resolution (e.g.,

~1deg) (Liao et al., 2022). The relationship between discharge (or drainage area) and river channel geometry is commonly used to determine the river width and depth in global river transport models (Xu et al., 2022b; Decharme et al., 2012). However, such relationship varies from watershed to watershed. It is not possible to use a single factor to derive the river geometry as it is affected by multiple factors such as discharge magnitude, seasonality, lithology, channel slope, etc.

To assess the transferability and robustness of our approach, we implemented the two-step calibration procedure in a

watershed located in a distinct climate and biome, specifically the Susquehanna River Basin. Figure S11 shows that the performance of simulating BFI, SWF, and streamflow seasonality are improved after the calibration with the proposed method.

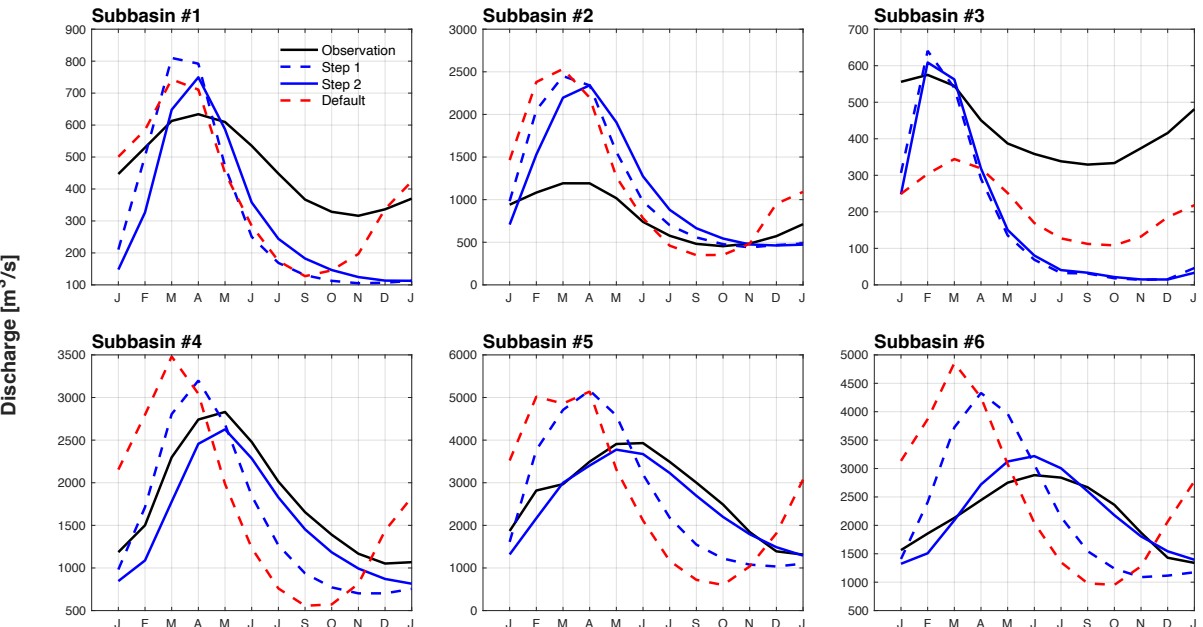

**Figure 12.** Simulated streamflow seasonality during 1981-2010 for the selected subbasins with default parameter values (red

dashed line) and calibrated parameter values after step 1 (blue dashed line) and step 2 (blue solid line). The black solid line denotes the observed streamflow seasonality.

## 4 Conclusion

In this study, we investigated the impacts of the runoff generation process and river routing process on streamflow variability in the coupled land and river configuration of E3SM using the Pantanal region as a case study. Previous studies

argued that floodplain storage effects are critical in capturing streamflow seasonality in global river transport models (Schrapffer et al., 2020; Yamazaki et al., 2011; Decharme et al., 2012), but we found that the impacts runoff generation process

on the streamflow variability should not be ignored. Calibrating either the runoff generation or river routing relevant parameters can improve the performance of simulated streamflow variability, but the system as a whole is not well captured, as the baseflow index, sensitivity of streamflow to climate change, water table depth, surface water dynamics remain highly biased. This implies that only calibrating one of the two processes results in overfitting and problematic parameter values, i.e., the impact of one process compensates for the bias resulted from the other process. Consequently, such calibration causes unrealistic simulation of other relevant variables. Therefore, a two-step calibration procedure is proposed for large-scale coupled land-river simulation to reconcile their impacts on streamflow variability. Specifically, the first step is to calibrate the hydrological parameters at the grid cell level with multiple selected objectives, followed by the second step to calibrate the river model at the basin level in terms of streamflow performance. The two-step calibration exhibits robust performance in the water cycle simulation. The sensitivity of streamflow to climate change was improved as well in our two-step calibration, suggesting the importance of including annual runoff trend in hydrological parameters calibration since trend analysis is of particular interest in Earth system modeling. The two-step calibration method demonstrates robust performance in calibrating E3SM coupled land-river configuration at a watershed with different climatology and biome.

Although we validated our two-step calibration method in an independent watershed (i.e., Susquehanna River basin), it remains unclear if it can improve the model performance in other watersheds with different characteristics and climatology. Additional evaluations and careful analysis are needed to apply the proposed calibration method to E3SM or other large scale hydrological models at global scales. Furthermore, there exist other challenges for the global calibration. First, streamflow observations are needed in the two-step calibration, but many watersheds at global scales are not gauged. Machine learning methods can be used to derive benchmark streamflow for the ungauged watersheds (Kratzert et al., 2019). Second, it is computationally expensive or even infeasible to run 2,000 global simulations for calibration. However, as demonstrated in Figure S9, a much smaller number of calibration simulations are needed to find approximately good parameter values. In conclusion, while different objective functions can be used in other model calibration, we suggest that both runoff generation and river routing processes should be carefully considered together to improve streamflow simulations with coupled land-river model.

**Code and Data Availability**

E3SMV2 is available from E3SM-Project website: https://github.com/E3SM-Project/E3SM (last access: Aug 2023) under BSD 3-clause license (https://github.com/E3SM-Project/E3SM/blob/master/LICENSE). The exact version of E3SM is published at https://github.com/donghuix/E3SM/releases/tag/v2-new-wetland-scheme (last access: Aug 2023), and is available at Zenodo: https://doi.org/10.5281/zenodo.6982264 (Xu, 2022). Instructions of running E3SMV2 can be found at: https://e3sm.org/model/running-e3sm/e3sm-quick-start/ (last access: Aug 2023). Matlab version R2019b Update 4 was used to pre-process datasets for simulations, post-process model outputs and plot results. The pre-processed datasets, Matlab pre-

and post-processing codes, and the script to run ELM-MOSART simulations are archived at Zenodo: https://doi.org/10.5281/zenodo.8290236 (Xu, 2023).

## Author contribution

D.X., G.B., and Z.T. designed the study. D.X. performed the simulations and analysis and wrote the initial manuscript under the mentorship of G.B.. T.Z. and H.L. advised D.X. on MOSART model. C.L., T.Z, H.L., and R.L. investigated the results. All authors contributed to the results discussion, review, and manuscript writing.

## Competing interests

The authors declare that they have no conflict of interest.

## Acknowledgments

This work was supported by the Earth System Model Development program area of the U.S. Department of Energy, Office of Science, Office of Biological and Environmental Research as part of the multi-program, collaborative integrated Coastal Modeling (ICoM) project. Chang Liao was supported through Next-Generation Ecosystem Experiments-Tropics, funded by the U.S. Department of Energy, Office of Science, Office of Biological and Environmental Research, at Pacific Northwest National Laboratory. The Pacific Northwest National Laboratory is operated by Battelle for the U.S. Department of Energy under Contract DE-AC05-76RLO1830.

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
