# Peer review of "Disentangling the Hydrological and Hydraulic Controls on Streamflow Variability in E3SM V2-A Case Study in the Pantanal Region"

_EGUsphere, 2023_

## Author Comment (AC1)

**General comments:**

I enjoyed reading this elaborated work on disentangling the hydrological and hydraulic controls on streamflow variability in E3SM V2. It is clearly written and organized, however missing some consistency in e.g. naming of experiments, and some figures could use a revision. Those adjustments in the text and figures would make it even more easy to follow the story in the text.

Xu et al. performed several experiments with the E3SM coupled with MOSART. The topic of studying parameter values of critical parameters of a land surface model is crucial, and especially the validation of model output against different observation products is important. It is a valuable contribution to the land surface modelling community.

The case study area of the Pantanal Region is interesting as the streamflow show a shift of about five months in the seasonality compared to precipitation.

**Response:** We thank the reviewer for the prompt review and constructive comments. We believe your comments are important and very helpful for improving the manuscript. In the revised manuscript, we will modify the experiment names, and update the figures as suggested. Please find our point-to-point response in the following.

I would like some more information about the calibration. I like the simplicity in the random sampling. Are 1000/2000 simulations an appropriate number? Why?

In my world calibration involves a mathematical optimizer, but your approach of doing several experiments with random sampling is an easy and simple approach to get an idea op optimal parameter values. It is not a demand, but maybe you could give some information on how the objective function distribute, and the performance of the "best" solution compared to the default? I would also like to know the parameter values of the "best" calibration. Those numbers would be valuable for other modelers.

**Response:** The performance of model with calibrated parameter values should increase as the number of calibration simulation increases. According to our previous calibration experience in E3SM (Xu et al., 2022), the three selected parameters in this study are the most sensitive model parameters for simulating runoff process. Randomly selecting 10 values on each parameter lead to a good coverage of the random variable space of the uncertainty parameters. For example, three parameters were selected for calibration in both ELM and MOSART, which leads to $10 \times 10 \times 10 = 1,000$ parameter combination in each calibration step. To further investigate if the 1,000 is enough for the calibration, we analyzed the calibrated model performance against the number of simulations used in the calibration. Figure R1a shows that the value of ELM's objective function (i.e., Step 1 calibration) decreases rapidly as the number of calibration simulations increases to 200 and the decrease in the objective function is much smaller for additional simulations. The initial set of 1,000 simulations was extended to include an additional 500 simulations with random parameters values, but those additional simulations did not further reduce the objective function significantly. This implies that the parameter calibrated with 1,000 simulations is a good approximation to the best parameter (e.g., the parameter calibrated with a much larger number of simulations). Similarly, the performance in step 2 (e.g., MOSART

calibration) did not increase after we increase the number of calibration simulations to be larger than ~100 (Figure R1b). The corresponding objective function and correlation coefficients from all the simulations of step 1 and step 2 can be found in the inserts of Figure R1. The two-step calibration significantly improves the model performance as the objection function with default parameter is 20.08 and the correlation coefficient with default simulated streamflow and observed streamflow is 0.2.

We will include the analysis for the number of calibrations in the revised manuscript. We will also add the calibrated parameter values in the supplementary material.

[Figure]

**Figure R1.** Change of objective function of step 1 calibration (left panel) and correlation between calibrated simulation and observation of step 2 calibration with number of calibration simulations. The histograms in left and right panels illustrates the distribution of the objective function and correlation coefficients from all the simulations during step 1 and step 2 calibration, respectively.

**Specific comments**

Page 7 line 169: please explain why it is an acceptable assumption.

**Response:** We made assumption that the annual runoff trend is spatially uniform within the watershed because there no reliable gridded runoff data to derive annual runoff trend at grid level. Although there are several subbasin gauges in the watershed, they have a shorter temporal coverage than the simulation period (i.e., 1979-2009). Therefore, the only way for us to include the trend in the objective function during the calibration is to assume the annual runoff trend is spatially uniform within the watershed. We will add a discussion of this limitation in the revised manuscript.

Figure 4: It is very hard to see difference in the size of the circles. The plot with rSD at xaxis: what is on the yaxis? Probably Manning coef., but this it not obvious. Please improve figure.

**Response:** Thanks for the suggestion. The reviewer is correct that the plot with rSD has Manning coefficient in the y-axis, which share the ticks with the colorbar. We will update Figure 4 in the revised manuscript to make it clearer.

Page 11 line 238: dosen´t it say some other numbers on figure?

**Response:** Thanks for catching the typo. We confirm the values reported in the figure is correct, and the value reported in the main text was from previous analysis. We will correct it in the revised manuscript.

Figure 6: I suggest to improve the readability of the figure (only a suggestion, not a need):

- (b): I am confused about what is on the left yaxis
- perhaps place the three plots with identical xaxis below each other. It would make it easy for the reader to get a quick overview.
- If there is no secondary yaxis, then always place the yaxis to the left.
- Be consistent about using the term observation and the actual name of the observation product in the legend

**Response**: Thanks for the suggestions. We believe your suggestions are very helpful to improve the readability of figure 6. In the revised manuscript, we will place the Y-axis to the left in subplot (b) and (d) in Figure 6 and we will change the legends in subplot (a), (c), and (d) to be "Gauge observation".

Page 13 line 270: *"The experiment of $f_{drain}$…"*. It would be easier to read and understand the text if the naming of the experiments were consistent. This apply to the whole paper.

**Response:** Thanks for the comment. In the revised manuscript, we will consistently use the naming of the experiments that we defined in Table 1 everywhere.

Figure 7: Please explain the term "ET trend" in the text. As I understand, you use the term "trend" regarding runoff in the paper. Please clarify in the text which trend you are referring to.

**Response:** In Figure 7, ET trend refers to the Sen's slope of annual basin averaged ET during the simulation period (i.e., 1979-2009). We will clarify it in the caption of Figure 7 in the revised manuscript.

Figure 8: There is something with the naming, why c and d?

- I suggest making the figure in the same way as figure 6, and with the same order of the plots.

**Response:** The figure subplot titles were messed up when we converted the figure format. We will correct it in the revised manuscript.

Figure 9: regarding legend in (a) and c): Be consistent with naming of the experiments in relation to text and other figures. The whole article would be much more readable if you were consistent with the naming.

**Response:** We appreciate this comment. We will modify the figure 9 in the revised manuscript to have consistent naming.

Page 16, line 320: There is no eq. 7

**Response:** Sorry for the confusion. It should be the objective function, i.e., Eq (4).

Figure 10: c): the dotted line is missing in legend

**Response:** The dotted line represents annual time series from observation. We didn't include it in the legend of Figure 10 because the Trend and p-value of the observation were reported in Figure 5. We found adding the observation in the legend will make Figure 10 busy. In the revised manuscript, we will describe the dotted line in the caption with relevant metrics.

**Reference**

Xu, D., Bisht, G., Sargsyan, K., Liao, C. and Leung, L.R. 2022. Using a surrogate-assisted Bayesian framework to calibrate the runoff-generation scheme in the Energy Exascale Earth System Model (E3SM) v1. Geosci. Model Dev. 15(12), 5021-5043.

---

## Author Comment (AC2)

**General comments**

Thank you for the well written and organized manuscript.

Calibration of parameters for hydrological and hydraulic processes is very important for river discharge, and I believe that calibration should be conducted carefully when applying to the global scale. It is difficult to apply the current results to the global scale. Further analysis of the current results is necessary.

**Response:** Thank you for reviewing our manuscript and providing the constructive comments for us to improve the manuscript. We agree calibration should be conducted carefully at global scales. Indeed, this work was inspired by current global calibration studies. As we argued in the manuscript, current calibration studies only focused on one process to improve the simulated streamflow but resulted in poor performance in other relevant processes. We aim to address this gap by proposing a two-step calibration method. In the current manuscript, we already tested our two-step calibration method in another watershed with different watershed characteristic and climatology (e.g., Susquehanna River basin locates in northeastern US). Without modification in the calibration procedure, the calibrated model shows improved performance in capturing baseflow index, surface water dynamics, streamflow variation, and annual streamflow trend. We note that the proposed two-step computational method is expensive to apply at global scales since it would take 2,000 global coupled ELM-MOSART simulations, which is beyond the scope of this study. In addition, another challenge for the calibration at global scales with the two-step method is that many basins are not gauged for monitoring streamflow. We will add a paragraph to discuss the challenges of applying our method to global scales in the revised manuscript. We think we need a follow up study in the future for the calibration, analysis, and evaluations at global scales.

Please find our point-to-point response to your comments in the following.

Although it may not be the purpose of this study, specifically, an analysis of the relationship between the parameters and the characteristics of the target river basin (precipitation, soil, topography/geology, etc.) would increase its applicability to the global scale.

**Response:** Many parameters in ELM and MOSART are derived from surface and subsurface conditions. For example, in ELM, saturated hydraulic conductivity and specific yield are estimated based on soil types, maximum drainage rate is determined by topographic slope, etc. In MOSART, river length and slope are derived from high resolution DEM. However, some other parameters should be determined based on sensitivity analysis and calibration, such as the parameters selected in this study. This is because ESM is typically too coarse to represent some physical processes, and empirical functions are used to parameterize those processes with simplifications. Specifically, $f_{drain}$ and $f_{over}$ are decay factors of the exponential function in the subsurface and surface runoff generation processes, respectively. According to the development of the runoff scheme in our model (e.g., simple TOPMODEL-based runoff parameterization), $f_{drain}$ and $f_{over}$ should be determined through sensitivity analysis or calibration against hydrograph recession curve (Niu et al., 2005; Niu et al., 2007). $f_c$ is a threshold below which no single connected inundated area spans the grid cell. It is used to quantify the fraction of the

inundated portion of the grid cell that is interconnected according to percolation theory. In other words, $f_c$ determines the maximum inundation extent, above which the water will outflow from the inundated area. Although the maximum inundated area should be controlled by topographic variation, high resolution data that captures the topographic variation under the inundated areas is not available.

There exist observations of river width and depth at very high spatial resolution. But it is challenging to upscale the observed river width and depth to ESM resolution (e.g., around 1deg) (Liao et al., 2022). The relationship between discharge (or drainage area) and river channel geometry is commonly used to determine the river width and depth. However, such relationship varies from watershed to watershed. It is not possible to use single factor to derive the river geometry as it is affected by multiple factors such as discharge magnitude, seasonality, lithology, channel slope, etc. Overall, the coarse spatial resolution of ESM and its simplifications in physical process make it hard to identify the relationship between selected parameters and watershed characteristic. In addition, according to our calibration experience in ESM, the most effective method to improve model performance and representation of the selected parameters is calibration.

Furthermore, we have difficulty in determining whether this two-step calibration is a good idea.The reason is that we do not know what kind of changes the first and second steps brought about in the river discharge, respectively.

Since this is a two-step calibration, there should be an answer for the first step, which can be shown in Fig. 12 to indicate how the river discharge changed from the default. It would be necessary to show how the river discharge changed and at what locations.

**Response:** Please find in the Figure R1 for the simulated streamflow after calibration of step 1 and step 2. Compared to the default simulation (red dashed line), the step 1 (blue dashed line). calibration shifts the streamflow by about 1 month. The Step 2 calibration (blue solid line) further delays the streamflow seasonality, especially for the downstream subbasins (subbasin #5 and #6).  We will update Figure 12 in the main text in the revision.

[Figure]

**Figure R1.** Simulated streamflow seasonality during 1981-2010 for the selected subbasins with default parameter values (red dashed line), calibrated ELM parameter values and default MOSART parameter values (blue dashed line), and calibrated ELM and MOSART parameter values (blue solid line). The black solid line denotes the observed streamflow seasonality. The blue dashed and solid lines represent the calibrated streamflow after Step 1 and Step 2 calibration, respectively.

**Specific comments**

How about describing the characteristics of the sub basin observation points in 2.2? Currently, the entire river basin is described with a focus on outlet points, but it would help to reinforce the last part of the discussion. In particular, SB#3 has a high river discharge for the area of the upstream river basin. It would be nice to have a comparison between precipitation and runoff height, even if it is shown on the supplement.

**Response:** Thanks for your suggestion. We will add some descriptions of the selected sub-basins in section 2.2. We note the reviewer is correct that SB#3 has a relatively high discharge for the area of the upstream river basin. As shown in Figure R2, the runoff coefficient in SB#3 is much larger than other sub-basins. This is because SB#3 locates in elevated area with steeper slope than other sub-basins.

[Figure]

**Figure R2.** Monthly averaged precipitation and discharge for the selected subbasins. C denotes runoff coefficient, which is the ratio of discharge to precipitation.

What is the reason why you chose these three components for multi object funcition in 2.5, you wrote that you didn't include streamflow variability in 3.5 because we didn't know how much it would be affected by runoff process only, but why did you include SWF? I think the SWF is also affected by river routine model, MOSART.

**Response:** We picked these three components in the multi-objective function because we think these are very critical metrics relevant to streamflow variability. We acknowledge other objective can be used for model calibration in other models, for example, some ESMs/LSMs may not have the surface water dynamics. We argue that both hydrological and hydraulic processes should be carefully calibrated together in the revised manuscript, however, the modelers could adopt other multi-objective functions based on the available processes in the calibrated models and theirs needs for the calibration.

In simulation, SWF is the sum of pluvial inundation (i.e., simulated by the land component, ELM) and fluvial inundation (i.e., simulated by the river component, MOSART). Therefore, SWF is affected by both runoff generation and river routing processes. In the first step calibration for the multi-objective function, we only calibrate ELM parameters, with MOSART parameters constant. Thus, the SWF in the multi-objective function is not affected by the routing process. We will add a clarification for this in the revised manuscript.

In the comparison between 3.4 and 3.5 (Fig. 9(a)), why is the river discharge lower in the two-step case than in the case where hydrology and hydraulics are calibrated separately?

**Response:** When hydrology is calibrated separately, only streamflow variability is considered in the objective function. However, in the two-step calibration, multi-objective function is used. The calibrated parameter corresponds to the smallest multi-objective function of Eq (4) is different from the calibrated parameter in calibrating hydrology and hydraulic separately as other variables were considered. We will update the text in the revised manuscript for clarity.

Regarding discussion 3.5, you list three factors for underestimation of river discharge, but I think we need to be sure that these three factors are really contributing to the problem. For example, if we consider average evapotranspiration, increase the area of the river basin by 5%, and increase precipitation, how much runoff will be generated and whether this runoff can represent an underestimation of the river discharge, we can check this at the order level. Alternatively, we could implement the results for other precipitation products.

**Response:** The simulated streamflow underestimates the observed streamflow by about 11%, which is $-17\ [mm/yr]$ in absolute value. The smaller contributing area can contribute to ~5% in the streamflow underestimation, assuming the runoff from the missing area is similar to the basin averaged runoff. The precipitation ensemble (e.g., based on the selected precipitation datasets) in Table S1) leads to the average annual precipitation during 1979-2009 to be $1,107-1,229\ [mm/yr]$. CRUNCEP forcing that used in our study has the average annual precipitation to be $1,156\ [mm/yr]$. Therefore, the long-term uncertainty of the precipitation in the used forcing is about $-73-49\ [mm/yr]$. Similarly, considering the two evapotranspiration (ET) references, the simulated ET uncertainty is about $54-154\ [mm/yr]$. Assuming the water storage doesn't change in the simulation period, the long-term averaged runoff is the difference between the long-term precipitation and evaporation. Therefore, an approximate estimate of runoff uncertainty from precipitation and evaporation is about $-227--5\ [mm/yr]$. Overall, the three factors can explain the underestimation.

We will add this additional analysis in the revised manuscript.

After that, I think it is necessary to consider the factors that lead to an overestimation of evapotranspiration.

**Response:** There are several attributions for the uncertainty of simulated evapotranspiration (ET). First, the uncertainty in the forcing can affect ET process. Specifically, the shortwave incoming radiation and air temperature in the used atmosphere forcing (i.e., CRUNCEP) is larger than atmosphere forcing of The Global Soil Wetness Project Phase 3 (GSWP3) (Figure R3). The higher shortwave incoming radiation and air temperature can both lead to higher evapotranspiration. The second factor is uncertainty in the surface dataset in ELM, for example, vegetation types and corresponding lead area index (LAI), which are a very sensitive surface conditions for ET process. The third uncertainty is due to parameterization. The ET parameterization is based on Monin-Obukhov similarity theory. The estimation of aerodynamic resistance and stomatal resistance commonly contain substantial uncertainty. Although the streamflow is significantly impacted by ET processes, it is challenging to calibrate ET process due to the lack of ET observation. We will add additional discussion of potential factors for the ET overestimation in the revised manuscript.

[Figure]

**Figure R3.** Comparison of (a). seasonal short waving incoming; (b). annual short waving incoming; (c). seasonal temperature; and (d) annual temperature between GSWP3 (blue solid line) and CRUNCEP (red dashed line).

Liao, C., Zhou, T., Xu, D., Barnes, R., Bisht, G., Li, H.-Y., Tan, Z., Tesfa, T., Duan, Z., Engwirda, D. and Leung, L.R.  2022.  Advances in hexagon mesh-based flow direction modeling. Adv Water Resour 160, 104099.

Niu, G.-Y., Yang, Z.-L., Dickinson, R.E. and Gulden, L.E.  2005.  A simple TOPMODEL-based runoff parameterization (SIMTOP) for use in global climate models. Journal of Geophysical Research: Atmospheres 110(D21).

Niu, G.-Y., Yang, Z.-L., Dickinson, R.E., Gulden, L.E. and Su, H.  2007.  Development of a simple groundwater model for use in climate models and evaluation with Gravity Recovery and Climate Experiment data. Journal of Geophysical Research: Atmospheres 112(D7).

---

## Author Response (AR1)

**Referee#1**

**General comments:**

I enjoyed reading this elaborated work on disentangling the hydrological and hydraulic controls on streamflow variability in E3SM V2. It is clearly written and organized, however missing some consistency in e.g. naming of experiments, and some figures could use a revision. Those adjustments in the text and figures would make it even more easy to follow the story in the text.

Xu et al. performed several experiments with the E3SM coupled with MOSART. The topic of studying parameter values of critical parameters of a land surface model is crucial, and especially the validation of model output against different observation products is important. It is a valuable contribution to the land surface modelling community.

The case study area of the Pantanal Region is interesting as the streamflow show a shift of about five months in the seasonality compared to precipitation.

**Response:** We thank the reviewer for the prompt review and constructive comments. We believe your comments are important and very helpful for improving the manuscript. In the revised manuscript, we referred the experiment names consistently throughout the manuscript, and updated the figures as suggested. Please find our point-to-point response in the following.

I would like some more information about the calibration. I like the simplicity in the random sampling. Are 1000/2000 simulations an appropriate number? Why?

In my world calibration involves a mathematical optimizer, but your approach of doing several experiments with random sampling is an easy and simple approach to get an idea op optimal parameter values. It is not a demand, but maybe you could give some information on how the objective function distribute, and the performance of the "best" solution compared to the default? I would also like to know the parameter values of the "best" calibration. Those numbers would be valuable for other modelers.

**Response:** The performance of model with calibrated parameter values should increase as the number of calibration simulation increases. According to our previous calibration experience in E3SM (Xu et al., 2022), the three selected parameters in this study are the most sensitive model parameters for simulating runoff process. Randomly selecting 10 values on each parameter lead to a good coverage of the random variable space of the uncertainty parameters. For example, three parameters were selected for calibration in both ELM and MOSART, which leads to $10 \times 10 \times 10 = 1,000$ parameter combination in each calibration step. To further investigate if the 1,000 simulations are enough for the calibration, we analyzed the calibrated model performance against the number of simulations used in the calibration. Figure R1a shows that the value of ELM's objective function (i.e., Step 1 calibration) decreases rapidly as the number of calibration simulations increases to 200 and the decrease in the objective function is much slower for additional simulations. The initial set of 1,000 simulations was extended to include an additional 500 simulations with random parameter values, but those additional simulations did not further reduce the objective function significantly. This implies that the parameter calibrated

with 1,000 simulations is a good approximation to the best parameter value (e.g., the parameter calibrated with a much larger number of simulations). Similarly, the performance in step 2 (e.g., MOSART calibration) did not increase after we increase the number of calibration simulations to be larger than ~100 (Figure R1b). The corresponding objective function and correlation coefficients from all the simulations of step 1 and step 2 can be found in the insets of Figure R1. The two-step calibration significantly improves the model performance as the objective function with default parameter is 20.08 and the correlation coefficient with default simulated streamflow and observed streamflow is 0.2.

We included the analysis of the number of calibration simulations in Figure S9 and Line 415 – Line 418 of the revised manuscript. We also added the calibrated parameter values in Figure S10 in the supplementary material.

[Figure]

**Figure R1.** Change of objective function of step 1 calibration (left panel) and correlation between calibrated simulation and observation of step 2 calibration with the number of calibration simulations. The histograms in the left and right panels illustrates the distribution of the objective function and correlation coefficients from all the simulations during step 1 and step 2 calibration, respectively.

**Specific comments**

Page 7 line 169: please explain why it is an acceptable assumption.

**Response:** We made an assumption that the annual runoff trend is spatially uniform within the watershed because there are no reliable gridded runoff data to derive the annual runoff trend at grid level. Although there are several subbasin gauges in the watershed, they have a shorter temporal coverage than the simulation period (i.e., 1979-2009). Therefore, the only way for us to include the trend in the objective function during the calibration is to assume the annual runoff trend is spatially uniform within the watershed. We added this limitation in the revised manuscript at Line 179 – Line 182.

Figure 4: It is very hard to see difference in the size of the circles. The plot with rSD at xaxis: what is on the yaxis? Probably Manning coef., but this it not obvious. Please improve figure.

**Response:** Thanks for the suggestion. We increased the figure size to make the difference of circle sizes more obvious. The reviewer is correct that the plot with rSD has Manning coefficient in the y-axis, which share the ticks with the colorbar. We updated Figure 4 in the revised manuscript to make it clearer.

Page 11 line 238: dosen´t it say some other numbers on figure?

**Response:** Thanks for catching the typo. We confirm the values reported in the figure are correct, and the value reported in the main text was from a previous analysis. We corrected it in the revised manuscript.

Figure 6: I suggest to improve the readability of the figure (only a suggestion, not a need):

- (b): I am confused about what is on the left yaxis
- perhaps place the three plots with identical xaxis below each other. It would make it easy for the reader to get a quick overview.
- If there is no secondary yaxis, then always place the yaxis to the left.
- Be consistent about using the term observation and the actual name of the observation product in the legend

**Response**: Thanks for the suggestions. We believe your suggestions are very helpful to improve the readability of figure 6. In the revised manuscript, we placed the Y-axis to the left in subplot (b) and (d) in Figure 6 and we changed the legends in subplot (a), (c), and (d) to be "Gauge observation".

Page 13 line 270: *"The experiment of $f_{drain}$…"*. It would be easier to read and understand the text if the naming of the experiments were consistent. This apply to the whole paper.

**Response:** Thanks for the comment. In the revised manuscript, we consistently used the naming of the experiments that we defined in Table 1 everywhere.

Figure 7: Please explain the term "ET trend" in the text. As I understand, you use the term "trend" regarding runoff in the paper. Please clarify in the text which trend you are referring to.

**Response:** In Figure 7, ET trend refers to the Sen's slope of annual basin averaged ET during the simulation period (i.e., 1979-2009). We clarified it in the caption of Figure 7 in the revised manuscript.

Figure 8: There is something with the naming, why c and d?

- I suggest making the figure in the same way as figure 6, and with the same order of the plots.

**Response:** The figure subplot titles were messed up when we converted the figure format. We corrected them in the revised manuscript.

Figure 9: regarding legend in (a) and c): Be consistent with naming of the experiments in relation to text and other figures. The whole article would be much more readable if you were consistent with the naming.

**Response:** We appreciate this comment. We modified figure 9 in the revised manuscript to have consistent naming with the main text and Table 1.

Page 16, line 320: There is no eq. 7

**Response:** Sorry for the confusion. It should be the objective function, i.e., Eq (4).

Figure 10: c): the dotted line is missing in legend

**Response:** The dotted line represents annual time series from observation. We didn't include it in the legend of Figure 10 because the Trend and p-value of the observation were reported in Figure 5. We found adding the observation in the legend will make Figure 10 busy. In the revised manuscript, we described the dotted line in the caption with relevant metrics.

**Referee#2**

**General comments**

Thank you for the well written and organized manuscript.

Calibration of parameters for hydrological and hydraulic processes is very important for river discharge, and I believe that calibration should be conducted carefully when applying to the global scale. It is difficult to apply the current results to the global scale. Further analysis of the current results is necessary.

**Response:** Thank you for reviewing our manuscript and providing the constructive comments for us to improve the manuscript. We agree calibration should be conducted carefully at global scales. Indeed, this work was inspired by current global calibration studies. As we argued in the manuscript, current calibration studies only focused on one process to improve the simulated streamflow but resulted in poor performance in other relevant processes. We aim to address this gap by proposing a two-step calibration method. In the current manuscript, we already tested our two-step calibration method in another watershed with different watershed characteristics and climatology (e.g., Susquehanna River basin located in northeastern US). Without modification in the calibration procedure, the calibrated model shows improved performance in capturing baseflow index, surface water dynamics, streamflow variation, and annual streamflow trend. We note that the proposed two-step computational method is expensive to apply at global scales since it would take 2,000 global coupled ELM-MOSART simulations, which is beyond the scope of this study. In addition, another challenge for the calibration at global scales with the two-step method is that many basins are not gauged for monitoring streamflow. We added a paragraph to discuss the challenges of applying our method to global scales in the revised manuscript at Line 472 – Line 480. We think we need a follow up study in the future for the calibration, analysis, and evaluations at global scales.

Please find our point-to-point response to your comments in the following.

Although it may not be the purpose of this study, specifically, an analysis of the relationship between the parameters and the characteristics of the target river basin (precipitation, soil, topography/geology, etc.) would increase its applicability to the global scale.

**Response:** Many parameters in ELM and MOSART are derived from surface and subsurface conditions. For example, in ELM, saturated hydraulic conductivity and specific yield are estimated based on soil types, maximum drainage rate is determined by topographic slope, etc. In MOSART, river length and slope are derived from high resolution DEM. However, some other parameters should be determined based on sensitivity analysis and calibration, such as the parameters selected in this study. This is because ESM resolution is typically too coarse to represent some physical processes, and empirical functions are used to parameterize those processes with simplifications. Specifically, $f_{drain}$ and $f_{over}$ are decay factors of the exponential function in the subsurface and surface runoff generation processes, respectively. According to the development of the runoff scheme in our model (e.g., simple TOPMODEL-based runoff parameterization), $f_{drain}$ and $f_{over}$ should be determined through sensitivity analysis or

calibration against hydrograph recession curve (Niu et al., 2005; Niu et al., 2007). $f_c$ is a threshold below which no single connected inundated area spans the grid cell. It is used to quantify the fraction of the inundated portion of the grid cell that is interconnected according to percolation theory. In other words, $f_c$ determines the maximum inundation extent, above which the water will outflow from the inundated area. Although the maximum inundated area should be controlled by topographic variation, high resolution data that captures the topographic variation under the inundated areas is not available.

There exist observations of river width and depth at very high spatial resolution. But it is challenging to upscale the observed river width and depth to ESM resolution (e.g., around 1deg) (Liao et al., 2022). The relationship between discharge (or drainage area) and river channel geometry is commonly used to determine the river width and depth. However, such relationship varies from watershed to watershed. It is not possible to use a single factor to derive the river geometry as it is affected by multiple factors such as discharge magnitude, seasonality, lithology, channel slope, etc. Overall, the coarse spatial resolution of ESM and its simplifications in physical process make it hard to identify the relationship between selected parameters and watershed characteristics. In addition, according to our calibration experience in ESM, the most effective method to improve model performance and representation of the selected parameters is calibration.

We added this discussion in the revised manuscript at Line 419 – Line 442.

Furthermore, we have difficulty in determining whether this two-step calibration is a good idea.The reason is that we do not know what kind of changes the first and second steps brought about in the river discharge, respectively.

Since this is a two-step calibration, there should be an answer for the first step, which can be shown in Fig. 12 to indicate how the river discharge changed from the default. It would be necessary to show how the river discharge changed and at what locations.

**Response:** We thank the review for this suggestion. Please find in Figure R2 the simulated streamflow after calibration step 1 and step 2. Compared to the default simulation (red dashed line), the step 1 (blue dashed line) calibration shifts the streamflow by about 1 month. The step 2 calibration (blue solid line) further delays the streamflow seasonality, especially for the downstream subbasins (subbasin #5 and #6). We updated Figure 12 in the main text and added some discussion on it at Line 407 – Line 409 in the revision.

[Figure]

**Figure R2.** Simulated streamflow seasonality during 1981-2010 for selected subbasins with default parameter values (red dashed line), calibrated ELM parameter values and default MOSART parameter values (blue dashed line), and calibrated ELM and MOSART parameter values (blue solid line). The black solid line denotes the observed streamflow seasonality. The blue dashed and solid lines represent the calibrated streamflow after Step 1 and Step 2 calibration, respectively.

**Specific comments**

How about describing the characteristics of the sub basin observation points in 2.2? Currently, the entire river basin is described with a focus on outlet points, but it would help to reinforce the last part of the discussion. In particular, SB#3 has a high river discharge for the area of the upstream river basin. It would be nice to have a comparison between precipitation and runoff height, even if it is shown on the supplement.

**Response:** Thanks for your suggestion. We have added some descriptions of the selected sub-basins in section 2.2. We note the reviewer is correct that SB#3 has a relatively high discharge for the area of the upstream river basin. As shown in Figure R3, the runoff coefficient in SB#3 is much larger than other sub-basins. This is because SB#3 is located in elevated area with steeper slope than other sub-basins.

[Figure]

**Figure R3.** Monthly averaged precipitation and discharge for the selected subbasins. C denotes runoff coefficient, which is the ratio of discharge to precipitation.

What is the reason why you chose these three components for multi object funcition in 2.5, you wrote that you didn't include streamflow variability in 3.5 because we didn't know how much it would be affected by runoff process only, but why did you include SWF? I think the SWF is also affected by river routine model, MOSART.

**Response:** We picked these three components in the multi-objective function because we think these are very critical metrics relevant to streamflow variability. We acknowledge other objective functions can be used for model calibration in other models; for example, some ESMs/LSMs may not have the surface water dynamics. We argue that both hydrological and hydraulic processes should be carefully calibrated together in the revised manuscript, however, the modelers may adopt other multi-objective functions based on the processes represented in the calibrated models and theirs needs for the calibration.

In the simulation, SWF is the sum of pluvial inundation (i.e., simulated by the land component, ELM) and fluvial inundation (i.e., simulated by the river component, MOSART). Therefore, SWF is affected by both runoff generation and river routing processes. In the first step calibration for the multi-objective function, we only calibrate ELM parameters, with the default MOSART parameter values. Thus, SWF in the multi-objective function is not affected by the routing process. We added this clarification in the revised manuscript at Line 172 – Line 175.

In the comparison between 3.4 and 3.5 (Fig. 9(a)), why is the river discharge lower in the two-step case than in the case where hydrology and hydraulics are calibrated separately?

**Response:** When hydrology is calibrated separately, only streamflow variability is considered in the objective function. However, in the two-step calibration, multi-objective function is used. The calibrated parameter corresponding to the smallest multi-objective function of Eq (4) is different from the calibrated parameter when hydrology and hydraulic are calibrated separately as other variables were considered.

Regarding discussion 3.5, you list three factors for underestimation of river discharge, but I think we need to be sure that these three factors are really contributing to the problem. For example, if we consider average evapotranspiration, increase the area of the river basin by 5%, and increase precipitation, how much runoff will be generated and whether this runoff can represent an underestimation of the river discharge, we can check this at the order level. Alternatively, we could implement the results for other precipitation products.

**Response:** The simulated streamflow underestimates the observed streamflow by about 21%, which is 32 $[mm/yr]$ in absolute value. The smaller contributing area can contribute to ~5% in the streamflow underestimation, assuming the runoff from the missing area is similar to the basin averaged runoff. The precipitation ensemble (e.g., based on the selected precipitation datasets) in Table S1) leads to the average annual precipitation during 1979-2009 to be 1,107– 1,229 $[mm/yr]$. The CRUNCEP forcing used in our study has an average annual precipitation of 1,156 $[mm/yr]$. Therefore, the long-term uncertainty of the precipitation in the forcing is about $-73 - 49$ $[mm/yr]$. Similarly, considering the two evapotranspiration (ET) references, the simulated ET uncertainty is about $54 - 154$ $[mm/yr]$. Assuming the water storage doesn't change in the simulation period, the long-term averaged runoff is the difference between the long-term precipitation and evaporation. Therefore, an approximate estimate of runoff uncertainty from precipitation and evaporation is about $-227 - -5$ $[mm/yr]$. Overall, the three factors can explain the underestimation.

We added this additional analysis at Line 374 – Line 388 in the revised manuscript.

After that, I think it is necessary to consider the factors that lead to an overestimation of evapotranspiration.

**Response:** There are several attributions for the uncertainty of simulated evapotranspiration (ET). First, the uncertainty in the forcing can affect ET process. Specifically, the shortwave incoming radiation and air temperature in the atmosphere forcing (i.e., CRUNCEP) are larger than atmosphere forcing of The Global Soil Wetness Project Phase 3 (GSWP3) (Figure R4). The higher shortwave incoming radiation and air temperature can both lead to higher evapotranspiration. The second factor is uncertainty in the surface dataset in ELM, for example, vegetation types and corresponding leaf area index (LAI), which are very sensitive surface conditions for ET process. The third uncertainty is due to parameterization. The ET parameterization is based on Monin-Obukhov similarity theory. The estimation of aerodynamic resistance and stomatal resistance commonly contain substantial uncertainty. Although the streamflow is significantly impacted by ET processes, it is challenging to calibrate ET process due to the lack of ET observation. We explained the potential factors for the ET overestimation at Line 383 and Line 387 – Line 388 in the revised manuscript.

[Figure]

**Figure R4.** Comparison of (a). seasonal short waving incoming; (b). annual short waving incoming; (c). seasonal temperature; and (d) annual temperature between GSWP3 (blue solid line) and CRUNCEP (red dashed line).

**Reference**

Liao, C., Zhou, T., Xu, D., Barnes, R., Bisht, G., Li, H.-Y., Tan, Z., Tesfa, T., Duan, Z., Engwirda, D. and Leung, L.R.  2022.  Advances in hexagon mesh-based flow direction modeling. Adv Water Resour 160, 104099.

Niu, G.-Y., Yang, Z.-L., Dickinson, R.E. and Gulden, L.E.  2005.  A simple TOPMODEL-based runoff parameterization (SIMTOP) for use in global climate models. Journal of Geophysical Research: Atmospheres 110(D21).

Niu, G.-Y., Yang, Z.-L., Dickinson, R.E., Gulden, L.E. and Su, H.  2007.  Development of a simple groundwater model for use in climate models and evaluation with Gravity Recovery and Climate Experiment data. Journal of Geophysical Research: Atmospheres 112(D7).

Xu, D., Bisht, G., Sargsyan, K., Liao, C. and Leung, L.R.  2022.  Using a surrogate-assisted Bayesian framework to calibrate the runoff-generation scheme in the Energy Exascale Earth System Model (E3SM) v1. Geosci. Model Dev. 15(12), 5021-5043.

---

## Author Response (AR2)

I think manuscript is well written. However, it is difficult to make a final judgment because the part regarding supplement in the manuscript is different from the supplement provided.

If you could provide a new supplement, I think it would be worthy of publication.

**Response:** We thank the reviewer for reviewing our revised manuscript. We found the corresponding author forgot to upload the revised supplementary materials file. In this revision, the revised materials file was correctly uploaded.

Extra captions remain in the figures in the manuscript(e.g., Figure 6(d)), which should also be corrected.

**Response:** Thank you for catching the errors. In the revised manuscript, we removed the extra captions in both Figure 6d and Figure 8a.